# Honor Among Bandits:
# No-Regret Learning for Online Fair Division

**Ariel D. Procaccia**[*]        **Benjamin Schiffer**[†]        **Shirley Zhang**[‡]

## Abstract

We consider the problem of online fair division of indivisible goods to players when there are a finite number of types of goods and player values are drawn from distributions with unknown means. Our goal is to maximize social welfare subject to allocating the goods fairly in expectation. When a player's value for an item is unknown at the time of allocation, we show that this problem reduces to a variant of (stochastic) multi-armed bandits, where there exists an arm for each player's value for each type of good. At each time step, we choose a distribution over arms which determines how the next item is allocated. We consider two sets of fairness constraints for this problem: envy-freeness in expectation and proportionality in expectation. Our main result is the design of an explore-then-commit algorithm that achieves $\tilde{O}(T^{2/3})$ regret while maintaining either fairness constraint. This result relies on unique properties fundamental to fair-division constraints that allow faster rates of learning, despite the restricted action space. We also prove a lower bound of $\tilde{\Omega}(T^{2/3})$ regret for our setting, showing that our results are tight.

## 1   Introduction

Fair allocation of indivisible goods is a fundamental problem with a wide range of applications; implemented algorithms for this task have been widely used in practice [14]. We consider the online fair division setting, which introduces additional complexities as items arrive one by one and each item must be immediately and irrevocably allocated at its time of arrival. Crucially, this allocation must be done without knowledge of future items [5]. One motivating example for this setting is a food bank that receives donations for a region and then allocates these donations among many different food pantries and soup kitchens in that region. Donations are often perishable, and therefore must be immediately allocated. Furthermore, donations can be unpredictable, and hence knowledge of future items is limited.

Two standard notions of fairness are *envy-freeness* and *proportionality*. Envy-freeness implies that every player is at least as happy with their own allocation as with any other player's allocation. Intuitively, envy-freeness guarantees that no player will want to trade their allocation for that of another player. Proportionality is a slightly weaker notion, which requires only that each of the $n$ players receive at least a $1/n$ fraction of their total value for all items. Finding a solution which is envy-free or proportional is often interesting in and of itself, as can be seen from many previous results in fair division [7, 30, 12, 3]. In cases where there may exist multiple envy-free or proportional allocations, however, a natural goal is then to find the best solution among such allocations [13]. In our work, we evaluate the quality of a fair solution by its (utilitarian) *social welfare*, which is defined as the sum over all players of each player's value for their own allocation.

We take a probabilistic approach to analyzing online fair division. In particular, we assume that there are a finite number of item types, and each player's value for each type of item is drawn from

---

[*]Paulson School of Engineering and Applied Sciences, Harvard University | *E-mail*: ariel-pro@seas.harvard.edu.

[†]Department of Statistics, Harvard University | *E-mail*: bschiffer1@g.harvard.edu.

[‡]Paulson School of Engineering and Applied Sciences, Harvard University | *E-mail*: szhang2@g.harvard.edu.

38th Conference on Neural Information Processing Systems (NeurIPS 2024).

a random distribution. In practice, these distributions would not be known in advance and must be learned as items are allocated. For example, consider again the food bank. When a new food pantry opens, the values of that food pantry for different types of products are unknown. After items have been allocated to the food pantry, however, the food bank can easily collect information on the demand for various item types at the food pantry. Therefore, we primarily consider the setting where the player distributions are unknown in advance, and a player's true value for an item is observed if and only if that player receives the item. This problem can be viewed as a variant of the multi-armed bandits problem, as the goal is to learn unknown distributions (player values) while maintaining high reward (social welfare), subject to fairness constraints; with a finite number of types of items, pulling an arm represents allocating a specific item type to a specific player.

As is standard in the multi-armed bandits literature, we use the notion of *regret* to measure the difference between our algorithm's performance and that of the optimal policy that knows the value distributions and is subject to the same fairness constraints. Our overarching challenge is this: *design online allocation algorithms that achieve low regret while maintaining fairness in the form of envy-freeness or proportionality.*

## 1.1 Our Results

Our main result is that there exists a simple optimization-based explore-then-commit algorithm that achieves $\tilde{O}(T^{2/3})$ regret and maintains envy-freeness in expectation (Algorithm 1 and Theorem 1). A variant of the same algorithm achieves $\tilde{O}(T^{2/3})$ regret while maintaining proportionality in expectation. The key step of the algorithm is a linear program-based optimization that guarantees that the constraints are satisfied without significantly decreasing social welfare.

The main difficulty in this learning problem is that the envy-freeness and proportionality constraints depend on the unknown value distributions and may be tight constraints without any slack. We therefore develop novel machinery that relies on fundamental properties of these fairness notions. One observation is that our fairness constraints are always satisfied when players are treated equally. Another crucial property is that when players have unequal values, these fairness notions can be satisfied with slack (Property 2). The latter property is especially challenging to show for envy-freeness, and the combinatorial algorithm that achieves it (Lemma 1) should be of independent interest to researchers in fair division.

## 1.2 Related Work

**Online Fair Division.** Work in online fair division generally deals with dividing goods when there is uncertainty about the future. Early work in the area focused on axiomatic questions [31, 1].

Our paper is most closely related to work by Benadè et al. [5]. Like us, they consider a setting where indivisible items arrive online and must be allocated immediately and irrevocably to players. They study several models for how the values of items are determined, ranging from a model where values are drawn i.i.d. from a distribution common to all players and items to, at the other extreme, an adversarial model with worst-case values. There are two fundamental differences between their work and ours. First, Benadè et al. [5] do not optimize social welfare; rather, they seek to either just minimize envy, or do so while (approximately) satisfying the axiomatic notion of Pareto efficiency. Second, and more crucially, they assume that the values of all players for an item are known at the time of its arrival, whereas in our model the values are unknown. It is precisely this modeling choice that induces a learning problem and underlies the connection between our setting and multi-armed bandits, which is absent in prior work in online fair division.

Like us, Yamada et al. [35] also study online fair division through the lens of bandit learning. Their setting is similar to ours in that they consider a finite number of item types where player values for item types are initially unknown. However, [35] do not guarantee fairness through constraints – instead, they incorporate fairness through their objective function of Nash social welfare. [35] also make an additional assumption that player values are normalized, which we do not require for our results.

**Fairness in Multi-armed Bandits.** The other main body of literature related to our paper is multi-armed bandits with constraints. One notion of fairness in multi-armed bandits is the idea that similar individuals and/or groups should be treated similarly [9, 18, 23]. The fairness constraint of Joseph et al. [17] is that a worse arm is not pulled with higher probability than a better arm. Their definition of fairness is actually incompatible with maintaining envy-freeness (or proportionality), because

maintaining envy-freeness may require allocating an item to a player with lower value to prevent envy. Another common fairness constraint in multi-armed bandits is that every arm receives a minimum fraction of pulls [10, 11, 20, 28]. This notion of fairness is also not compatible with envy-freeness because the optimal envy-free allocation may never give a player a specific item type. There also exist many other fairness notions in contextual bandits that are farther from our setting [15, 29, 32, 34]. Wei et al. [33] analyze a form of envy-freeness in contextual bandits, but their envy-freeness notion depends only on the treatment probabilities instead of the values.

Our paper is also closely related to work on multi-armed bandits subject to general linear constraints. Multiple works in linear bandits study "safety" with respect to a linear constraint that depends on the unknown true mean values [2, 8, 26]. Amani et al. [2] focus on a single constraint and specifically show that if there is positive slack in the optimal solution, then $\tilde{O}(T^{1/2})$ regret is possible. If there is zero slack, however, their algorithm only achieves $\tilde{O}(T^{2/3})$ regret. This differs from our work because envy-freeness and proportionality involve multiple constraints that can have zero slack. Note that our setting is similar but not equivalent to linear bandits, as a single arm is pulled in each step in our setting. There also exist many results for cumulative constraints in bandits [21, 22]. These are less closely related to our model as we consider constraints that must hold at every time step. Finally, there is a branch of multi-armed bandits that studies constraints in expectation at each step as in our paper. However, these works are also in the linear bandits setting and again require a safety gap that fairness constraints such as envy-freeness may not guarantee [27].

**Practical Motivation.** Mertzanidis et al. [25] apply online fair division algorithms through a partnership with a program in Indiana that redistributes rejected truckloads of food. The program, known as Food Drop, allocates 10,000+ pounds of rejected food per month to food banks. In this application, the available food arrives in an online and unpredictable way, and the trucks must be allocated immediately. More generally, the specific food donations depend on what items grocery stores or restaurants have remaining at the end of the day. Therefore, donations are unpredictable, which we model through randomness.

In practice, utilities for food donations such as in the Food Drop program may not be additive. However, if the deliveries are sufficiently infrequent, then additive player utilities are likely to be a good approximation. For example, in the food allocation data of [19], there were a total of 1760 donations from 169 donors over the course of five months and 277 organizations that received donations. Therefore, the organizations receive donations every 3-4 weeks on average, suggesting that donations can be largely regarded as independent.

## 2 Model

In this section we introduce our basic setting and terminology.

### 2.1 Online Allocation With Unknown Values

Suppose we have a set of players $N = [n]$ and a set of object types $M = [m]$. Given a set of $T$ indivisible items, an *allocation* $A = (A_1, ..., A_n)$ is a partition of the $T$ items among the $n$ players, where player $i$ receives the items in $A_i$. In our model, we assume that every item $j$ has a type $k(j) \in [m]$, and that there exists a (possibly unknown) matrix $\mu^*$ such that each player $i$'s value for an item of type $k \in [m]$ is independently drawn from a sub-Gaussian distribution with mean $\mu_{ik}^*$. Player values are assumed to be independent across both players and items. We will often refer to $\mu_i^*$ as the vector of mean values for player $i$. For a specific item $j$, we denote player $i$'s value for item $j$ as $V_i(j)$, and similarly, player $i$'s value for their allocation $A_i$ as $V_i(A_i) = \sum_{j \in A_i} V_i(j)$. The (utilitarian) *social welfare* of an allocation $A$ is $\mathrm{sw}(A) = \sum_{i=1}^{n} V_i(A_i)$.

We consider algorithms in the following online setting. At each time step $t \in [T]$, an item $j_t$ of type $k_t$ arrives, where $k_t \sim \mathcal{D}$, for some known distribution $\mathcal{D}$ supported on $[m]$. We will assume that $\mathcal{D} = \mathrm{Unif}([m])$, or in other words that every item has an equal probability of being type $1, ..., m$. We make this choice purely for ease of exposition, in order to simplify notation; our results and techniques extend seamlessly to arbitrary distributions $\mathcal{D}$ that do not depend on $T$, as we explain in Appendix C.1. The algorithm observes the item type $k_t$, and must then immediately allocate the item $j_t$ to a player $i_t$, at which time the algorithm observes that player's value $V_{i_t}(j_t)$. Note that the algorithm does not observe any other player's value for item $j_t$. The high-level goal is to allocate these items in a manner that maximizes the social welfare of the final allocation of all $T$ items.

We denote $X \in \mathbb{R}^{n \times m}$ as a valid fractional allocation if $\sum_i X_{ik} = 1$ for every $k \in [m]$. One valid fractional allocation we will often refer to is the *uniform at random* allocation (UAR), where every entry is $\frac{1}{n}$. At every time-step $t$, before observing $k_t$, the allocation algorithm ALG takes as input the history $H_t = \{(k_{t'}, i_{t'}, V_{i_{t'}}(j_{t'})) : t' < t\}$ and returns a fractional allocation $X_t = \text{ALG}(H_t)$, where $(X_t)_{ik}$ represents the probability of allocating the item to player $i$ if the item is of type $k$. If the next item is of type $k_t$, then the algorithm allocates the item randomly among the $n$ players according to the distribution induced by the $k_t^{\text{th}}$ column of $X_t$, i.e. $(X_t^\top)_{k_t}$. Therefore, the $t^{\text{th}}$ item is allocated to player $i$ with probability $(X_t)_{ik_t}$. We denote the final realized allocation that ALG returns as $A(\text{ALG})$, and the corresponding partial allocation up to time $\tau$ as $A^\tau(\text{ALG})$. This online process is summarized in pseudo-code in Appendix A.

We will also assume (explicitly in our theorem statements) that for all $i, k$, there exist known constants $a, b > 0$ such that $a \le \mu_{ik}^* \le b$. This assumption is necessary because if we allow the means of values to be arbitrary close to zero, then it can be impossible to achieve regret of $o(T)$. This is formalized in Theorem 11 in Appendix C.2.

## 2.2 Fairness Notions

We will primarily use two metrics of fairness to evaluate an online allocation algorithm ALG: envy-freeness in expectation and proportionality in expectation. Both are defined below. For two vectors $x, y \in \mathbb{R}^n$, we use $\langle x, y \rangle = x \cdot y$ to represent the dot product of the two vectors.

**Definition 1.** Let $X_t = \text{ALG}(H_t)$ be the fractional allocation used by algorithm ALG at time $t$ given history $H_t$. Then ALG satisfies *envy-freeness in expectation (*EFE*)* if for all $t$ and all $H_t$, $(X_t)_i \cdot \mu_i^* \ge \max_{i' \in [n]} (X_t)_{i'} \cdot \mu_i^*$ for all $i$.

**Definition 2.** Let $X_t = \text{ALG}(H_t)$ be the fractional allocation used by algorithm ALG at time $t$ given history $H_t$. Then ALG satisfies *proportionality in expectation (*PE*)* if for all $t$ and all histories $H_t$, $(X_t)_i \cdot \mu_i^* \ge \frac{1}{n} \sum_{i' \in [n]} (X_t)_{i'} \cdot \mu_i^*$ for all $i$.

Intuitively, envy-freeness in expectation is equivalent to maintaining that at every time step $t$ and before observing the item type $k_t$, no player prefers the fractional allocation of any other player in $X_t$. Similarly, proportionality in expectation is equivalent to maintaining that at every time step $t$ and before observing the item type $k_t$, the expected value of every player for their fractional allocation is at least $1/n$ times that player's value if they received the item with probability 1.

In Appendix B, we justify some of the implicit choices behind these definitions. Specifically, we discuss why we consider envy-freeness in expectation rather than its realization, and also why we require envy-freeness in expectation to hold at every individual time step. Analogous results for proportionality can be found in Appendix B.1. For the former question, Theorem 3 shows that in our setting, no algorithm can with high probability output an allocation $A(\text{ALG})$ with realized envy less than $\sqrt{T}$. Note that Benadè et al. [5] show that in the adversarial setting, no algorithm can guarantee $o(\sqrt{T})$ realized envy. Conversely, they also show that when values are generated randomly and observed before allocation, there exists an algorithm that *can* guarantee $o(1)$ realized envy with high probability. Theorem 3 shows that when values are still generated randomly but are *unknown* at the time of allocation (as in our setting), no algorithm can guarantee $o(\sqrt{T})$ realized envy with high probability. We complement Theorem 3 with Theorem 4, which shows that any algorithm ALG that satisfies envy-freeness in expectation will output a final allocation $A(\text{ALG})$ with realized envy of at most $\sqrt{T} \log(T)$ with high probability. Therefore, envy-free in expectation algorithms are within a $\log(T)$ factor of being "optimal" in terms of final realized envy.

We also show that requiring envy-freeness in expectation at every time step does not lead to any social welfare loss compared to requiring envy-freeness in expectation only at the end of $T$ rounds. More specifically, Theorem 5 (again in Appendix B) implies that requiring that no player is envious in expectation of any other player at the end of all $T$ rounds is equivalent to maintaining envy-freeness in expectation at all times $t \in [T]$ when maximizing social welfare. A key step of our proof of Theorem 5 is showing that for every time- or history-dependent algorithm ALG which achieves envy-freeness in expectation at the end of $T$ rounds, there exists another algorithm $\text{ALG}'$ that is time- and history-independent, envy-free in expectation at every time step, and achieves the same social welfare. Therefore, maximizing social welfare only over algorithms which are envy-free in expectation at every time step is sufficient even if envy-freeness in expectation at the end of $T$ rounds is all that is desired.

We can formulate our fairness notions as linear constraints, in the spirit of prior work in fair division [4]. Formally, define $\langle A, B \rangle_F$ as the Frobenius inner product of matrices $A$ and $B$. For $B \in \mathbb{R}^{n \times m}, c \in \mathbb{R}$, and a fractional allocation $X$, we represent the linear constraint $\langle B, X \rangle_F \geq c$ as the tuple $(B, c)$. A fractional allocation $X$ satisfies a set of $L$ linear constraints $\{(B_\ell, c_\ell)\}_{\ell=1}^L$ if for all $\ell \in [L]$, $\langle B_\ell, X \rangle_F \geq c_\ell$. Because the constraints represent "fairness in expectation" relative to the mean values, we will explicitly let the constraint matrix $B_\ell(\mu^*)$ be a function of the mean value matrix $\mu^*$. Therefore, we will consider sets of constraints of the form $\{(B_\ell(\mu^*), c_\ell)\}_{\ell=1}^L$. Because these constraints are functions of $\mu$, we will also refer to families of constraints $\left\{\{(B_\ell(\mu), c_\ell)\}_{\ell=1}^L\right\}_\mu$.

The following two remarks show how envy-freeness in expectation and proportionality in expectation can be represented within this framework.

**Remark 1.** For every $\ell \in [n^2]$, construct $B_\ell^{\mathrm{efe}}(\mu^*)$ as follows. Define $i = \lceil \frac{\ell}{n} \rceil$ and $i' = (\ell \mod n) + 1$. For every $k \in [K]$, let $(B_\ell^{\mathrm{efe}}(\mu^*))_{ik} = \mu_{ik}^*$ and $(B_\ell^{\mathrm{efe}}(\mu^*))_{i'k} = -\mu_{ik}^*$. For all $i'' \notin \{i, i'\}$, let $(B_\ell^{\mathrm{efe}}(\mu^*))_{i''} = 0$. Then the envy-freeness in expectation constraints for mean $\mu^*$ as defined in Definition 1 correspond to $\mathrm{efe}(\mu^*) := \{(B_\ell^{\mathrm{efe}}(\mu^*), 0)\}_{\ell=1}^{n^2}$.

**Remark 2.** For every $\ell \in [n]$, construct $B_\ell^{pe}(\mu^*)$ as follows. For every $k \in [m]$, $(B_\ell^{pe}(\mu^*))_{\ell k} = \frac{(n-1)}{n} \cdot \mu_{\ell k}^*$ and $(B_\ell^{pe}(\mu^*))_{ik} = -\frac{1}{n} \cdot \mu_{\ell k}^*$ for every $i \neq \ell$. Then the proportionality in expectation constraints for mean $\mu^*$ as defined in Definition 2 correspond to $\mathrm{pe}(\mu^*) = \{(B_\ell^{pe}(\mu^*), 0)\}_{\ell=1}^n$.

### 2.3 Regret and Problem Formulation

An algorithm ALG satisfies constraints $\{(B_\ell(\mu^*), c_\ell)\}_{\ell=1}^L$ if for all $t \in [T]$, the fractional allocation $X_t$ used by ALG at time $t$ satisfies the constraints $\{(B_\ell(\mu^*), c_\ell)\}_{\ell=1}^L$. When $\mu^*$ is known, the expected social welfare can be directly optimized over all algorithms ALG that satisfy constraints $\{(B_\ell(\mu^*), c_\ell)\}_{\ell=1}^L$. This problem is equivalent to solving LP (1) with $\mu = \mu^*$ and using the solution $Y^{\mu^*}$ as the fractional allocation for all time steps.

$$Y^\mu := \arg\max \ \langle X, \mu \rangle_F$$
$$\text{s.t. } \langle B_\ell(\mu), X \rangle_F \geq c_\ell \quad \forall \ell$$
$$\sum_i X_{ik} = 1 \quad \forall k \tag{1}$$

When $\mu^*$ is unknown, we define the regret of an algorithm ALG as follows. Note that the baseline algorithm in this definition of regret is the optimal allocation algorithm under the constraints when $\mu^*$ is known.

**Definition 3.** Let $Y^{\mu^*}$ be the solution to LP (1) for $\mu = \mu^*$. Let $X_t = \mathrm{ALG}(H_t)$ be the fractional allocation used by algorithm ALG at time $t$ given history $H_t$. Then the $T$-step regret of ALG for constraints $\{(B_\ell(\mu^*), c_\ell)\}_{\ell=1}^L$ is $T \cdot \langle Y^{\mu^*}, \mu^* \rangle_F - \sum_{t=0}^{T-1} \langle X_t, \mu^* \rangle_F$.

We are now ready to present the formal problem statement. Because the constraints depend on the unknown values that are being learned, we only require constraint satisfaction with high probability.

**Problem 1.** *Suppose we are given $n, m, T, a, b$ such that $0 < a \leq \mu_{ik}^* \leq b$ for all $i \in [n]$, $k \in [m]$. Given a family of fairness constraints $\left\{\{(B_\ell(\mu), c_\ell)\}_{\ell=1}^L\right\}_\mu$ representing either envy-freeness in expectation or proportionality in expectation, the goal is to construct an algorithm ALG such that with probability $1 - 1/T$, $X_t = \mathrm{ALG}(H_t)$ satisfies the constraints $(B_\ell(\mu^*), c_\ell)\}_{\ell=1}^L$ for all $t \in [T]$ and the regret of ALG for constraints $(B_\ell(\mu^*), c_\ell)\}_{\ell=1}^L$ is $o(T)$.*

Note that the $o(T)$ regret in Problem 1 will be $\tilde{O}(T^{2/3})$ for our results. We use the standard $O()$ and $\tilde{O}()$ notation with respect to the number of time steps $T$, and therefore the constants represented by this notation may depend on other problem parameters such as $n$ and $m$.

## 3 Fairness Machinery

Our goal in this section is to establish novel, fundamental properties of envy-freeness and proportionality in expectation, which will serve as a crucial part of the machinery underlying our regret bounds.

In the context of fairness, a natural assumption is that if a group of individuals are treated equally, then that group is considered to be treated fairly. In that spirit, our first key property is as follows.

**Property 1.** *For any $\ell \in [L]$, suppose that a fractional allocation $X \in \mathbb{R}^{n \times m}$ satisfies $X_{i_1} = X_{i_2}, \forall i_1, i_2 \in \{i : B_\ell(\mu)_i \neq \boldsymbol{0}\}$. Then $\langle B_\ell(\mu), X \rangle_F \geq c_\ell$.*

Informally, a set of constraints satisfies Property 1 if for any constraint in the set, the constraint is always satisfied when all players involved in the constraint have the same fractional allocation. An important consequence of Property 1 is that the uniform at random allocation satisfies every constraint. This implies that even with no information about the players' mean values, the uniform at random allocation will always be fair.

Note that envy-freeness in expectation satisfies Property 1 because any two players with equal allocations are never envious of each other. Proportionality in expectation also satisfies Property 1, because if every player has the same allocation, then every player is receiving exactly their proportional allocation.

**Observation 1.** *The envy-freeness in expectation and proportionality in expectation constraints satisfy Property 1.*

Our second key property is more technical and novel. Intuitively, the property requires the existence of a fractional allocation $X'$ that is only slightly worse than the optimal constrained allocation $Y^\mu$, but unlike the latter allocation, in $X'$ the constraints either hold with slack or all players involved in the constraint are treated equally. Interestingly, this property does not hold for arbitrary sets of linear constraints, but relies on structure inherent to the envy-freeness in expectation and proportionality in expectation constraints. The bulk of the theoretical work of this paper is proving that the envy-freeness in expectation and proportionality in expectation constraints satisfy this property.

**Property 2.** *Let $Y^\mu$ be the solution to LP (1). Then there exists constants (relative to $T$) $\gamma_0$ and $C_{P2} > 0$ such that for any $\gamma < \gamma_0$ and any $\mu$, there exists a fractional allocation $X'$ such that $\langle X', \mu \rangle_F \geq \langle Y^\mu, \mu \rangle_F - C_{P2}\gamma$, and such that for each $\ell \in [L]$, either*

1. *$\langle B_\ell(\mu), X' \rangle_F \geq c_\ell + \gamma$ or*
2. *$\forall i_1, i_2 \in \{i : B_\ell(\mu)_i \neq \boldsymbol{0}\}, X'_{i_1} = X'_{i_2}$.*

**Lemma 1.** *The family of envy-freeness in expectation constraints satisfies Property 2.*

*Proof sketch.* We will informally argue that we can transform $Y^\mu$ into a fractional allocation $X'$ which satisfies Property 2 through Algorithm 3. Algorithm 3 iterates over 'envy-with-slack-$\alpha$' graphs, which track whether a player prefers their allocation by at least $\alpha$ over another player's allocation. More specifically, given parameters $\mu, X,$ and $\alpha$, the corresponding 'envy-with-slack' graph has vertices $N$ and edge set $E$ such that a directed edge from $i$ to $i'$ exists if and only if $X_i \cdot \mu_i - X_{i'} \cdot \mu_i < \alpha$. The weight of each such edge is $X_i \cdot \mu_i - X_{i'} \cdot \mu_i$. At a high level, Algorithm 3 constructs 'envy-with-slack-$\alpha$' graphs with progressively smaller $\alpha$, with $\alpha \geq \gamma$ for all iterations. The algorithm operates on sets of nodes called *equivalence classes*, where every pair of nodes in an equivalence class has the same allocation. Algorithm 3 makes progress in every iteration by either 1) merging two equivalence classes, or 2) removing an edge from the graph.

An overview of the algorithm is as follows. In each iteration, Algorithm 3 generally performs one of three operations and decreases $\alpha$. First, if there exists an equivalence class $S$ with at least one incoming edge but no outgoing edges, then operation remove-incoming-edge transfers allocation probability from nodes in $S$ to all other nodes. This will remove all incoming edges to $S$. If such an equivalence class does not exist, then Algorithm 3 finds a special type of directed cycle in the 'envy-with-slack' graph. The directed cycle is chosen so that the outgoing edge of each node $i$ in the cycle is among $i$'s outgoing edges with minimal weight. Therefore, each node $i$ in the cycle is pointing to an $i' \in N$ for whom $i$ has the least slack. If there exists some node $i^* \in N$ which has an edge to some but not all of the nodes in the cycle, then operation cycle-shift gives each node in the cycle half of its current allocation and half of the next node's allocation. This will remove an outgoing edge from $i^*$. If such a node does not exist, then Algorithm 3 either decreases $\alpha$ to remove an edge or creates a new equivalence class by merging all equivalence classes that the nodes in the cycle belong to via operation average-clique.

However, such a merge may lead to envy, which is removed by Algorithm 4. Each call to Algorithm 4 removes envy from at least one edge. Algorithm 4 does so by first carefully redistributing allocation among the nodes until there exists a cycle where each node has non-negative envy (which is equivalent to a cycle with non-positive slack). Each node in the cycle is then given the allocation of the next

node in the cycle. We prove that each call to Algorithm 4 decreases the number of edges with envy, while not increasing the number of edges in the 'envy-with-slack' graph. Furthermore, Algorithm 4 does not significantly decrease the social welfare of the allocation.

The three operations and Algorithm 4 each take as input an allocation $X$ and returns a new allocation $X'$ which is close in social welfare to $X$. Furthermore, each iteration begins with an envy-free allocation, and the size of the edge set of the 'envy-with-slack' graphs never increases throughout the algorithm. The maximum size of an equivalence class is $n$, so an edge must be removed from the 'envy-with-slack' graph every $n$ iterations. There are at most $n^2$ edges, and the algorithm therefore terminates in at most $n^3$ iterations with an allocation which satisfies Property 2. For the numerous details, see Appendix F. □

**Lemma 2.** *The family of proportionality in expectation constraints satisfies Property 2.*

*Proof sketch.* Define the slack $S_i$ of a player $i$ for an allocation as the amount by which that player's value for their allocation is greater than their proportional value. In other words, the slack is the amount of welfare a player can lose and still satisfy the proportionality constraint. We construct the fractional allocation $X'$ in one of two different ways depending on the amount of total slack for the allocation $Y^\mu$ across all $n$ players.

If the amount of total slack across all $n$ players is less than $\frac{bn\gamma}{a}$, then we take $X' = \text{UAR}$. Note that the total slack is equivalent to the change in social welfare between $Y^\mu$ and UAR. Therefore, because the total slack was less than $\frac{bn\gamma}{a}$, the difference in social welfare between $Y^\mu$ and UAR is at most $\frac{bn\gamma}{a}$ which is $O(\gamma)$. Furthermore, by definition the UAR allocation satisfies option 2 of Property 2 for all constraints $\ell$.

If the amount of total slack is greater than $\frac{bn\gamma}{a}$, then we construct $X'$ from $Y^\mu$ by transferring allocation away from players with slack greater than the required $\gamma$ and redistributing this allocation so that every player has slack of at least $\gamma$. Specifically, each player $i$ loses $\Delta_{ik}$ of their allocation for item $k$, where

$$\Delta_{ik} := \frac{Y_{ik}^\mu}{\sum_{k'=1}^m Y_{ik'}^\mu} \cdot \frac{S_i}{\sum_{i'=1}^n S_{i'}} \cdot \frac{n\gamma}{a}.$$

The allocation $X'$ is then constructed as

$$X'_{ik} := Y_{ik}^\mu - \Delta_{ik} + \frac{1}{n} \sum_{i'=1}^n \Delta_{i'k}. \tag{2}$$

Intuitively, this can be viewed as each player putting a part of their allocation (proportional to $S_i \cdot \gamma$) into a communal "pot." The pot, consisting of $\sum_{i'=1}^n \Delta_{i'k}$ for item $k$, is then divided evenly among all $n$ players to form $X'$. By construction, no player loses more than $S_i$ social welfare when the pot is created, and every player receives at least $\gamma$ additional social welfare when the pot is redistributed. Therefore, in the resulting allocation $X'$, every player prefers their allocation to their proportional value by at least $\gamma$, i.e. each player has a slack of at least $\gamma$ for $X'$. Furthermore, the total difference in social welfare between $Y^\mu$ and $X'$ is at most $O(\gamma)$. We have thus shown that in both cases, $X'$ will satisfy all of the desired conditions. The full proof is relegated to Appendix E. □

It will be useful to introduce two further properties that are immediately satisfied by the definitions of envy-freeness in expectation and proportionality in expectation. Property 3 guarantees a form of Lipschitz continuity in $\mu$ for the constraints. This is unsurprising, as the entries in the matrices $B_\ell(\mu)$ for both envy-freeness in expectation and proportionality in expectation are linear in the entries of $\mu$. Property 4 guarantees that the non-zero entries in the constraint matrices stay the same for all values of $\mu$, which follows directly from Remarks 1 and 2 and the fact that $\mu_{ik}$ is bounded away from 0.

**Property 3.** *There exists a $K > 0$ such that $\forall \mu^1, \mu^2 \in [a,b]^{n \times m}$ and $\forall \epsilon > 0$, if $\|\mu^1 - \mu^2\|_1 \le \epsilon$ then $\|B_\ell(\mu^1) - B_\ell(\mu^2)\|_1 \le K\epsilon$.*

**Observation 2.** *The envy-freeness in expectation and proportionality in expectation constraints satisfy Property 3.*

**Property 4.** *For all $\mu, \mu' \in [a,b]^{n \times m}$, $\{i : B_\ell(\mu)_i \ne \boldsymbol{0}\} = \{i : B_\ell(\mu')_i \ne \boldsymbol{0}\}$.*

**Observation 3.** *The envy-freeness in expectation and proportionality in expectation constraints satisfy Property 4.*

Recall that Property 2 implies that for every constraint $\ell$, either the constraint $\ell$ has a slack of at least $\gamma$ for $X'$ or every player involved in constraint $\ell$ is treated equally under allocation $X'$. A slack of $\gamma$ in the constraint guarantees constraint satisfaction for all $\mu'$ close to $\mu$ if the constraints are continuous in $\mu$. Treating every player equally for a given constraint also guarantees that the constraints are satisfied for all $\mu'$ by Property 1. Therefore, Properties 1 and 2 together with continuity (Property 3) imply that there exists a fractional allocation $X'$ such that the social welfare of $X'$ is close to the social welfare of $Y^\mu$ and such that $X'$ not only satisfies the constraints for $\mu$, but also satisfies the constraints for any $\mu'$ close to $\mu$.

## 4  Algorithm and Regret Bounds

In this section, we present our main result, an explore-then-commit algorithm which achieves $\tilde{O}(T^{2/3})$ regret while maintaining either proportionality in expectation or envy-freeness in expectation. The key step in Algorithm 1 is the optimization in LP (3) to guarantee that the fairness constraints are satisfied with high probability. For $\mu \in \mathbb{R}_+^{n \times m}$ and $\epsilon \in \mathbb{R}_+^{n \times m}$, we define the confidence region $\mu \pm \epsilon = \{\mu' \in \mathbb{R}^{n \times m} : \mu_{ik} - \epsilon_{ik} \leq \mu'_{ik} \leq \mu_{ik} + \epsilon_{ik} \, \forall i, k\}$. Note that Algorithm 1 requires solving LPs with an infinite number of constraints, which we discuss further in Section 6.

---

**Algorithm 1** Fair Explore-Then-Commit

**Require:** $n, m, T, \{\{(B_\ell(\mu), c_\ell)\}_{\ell=1}^L\}_\mu$

  **for** $t \leftarrow 1$ to $T^{2/3} - 1$ **do**
    At time $t$, use fractional allocation $X^t = \text{UAR}$.
  **end for**
  $N_{ik} \leftarrow \sum_{\tau=0}^{T^{2/3}-1} \mathbb{1}_{k_\tau=k, i_\tau=i}$
  $\hat{\mu}_{ik} \leftarrow \frac{1}{N_{ik}} \sum_{\tau=0}^{T^{2/3}-1} \mathbb{1}_{k_\tau=k, i_\tau=i} V_{i_\tau}(j_\tau)$
  $\epsilon_{ik} = \sqrt{\log^2(4Tnm)/(2N_{ik})}$
  $\hat{X} \leftarrow$ Solution to the following LP:

$$\max_X \langle X, \hat{\mu} \rangle_F$$
$$\text{s.t.} \ \langle B_\ell(\mu), X \rangle_F \geq c_\ell \quad \forall \ell \in [L], \forall \mu \in \hat{\mu} \pm \epsilon$$
$$\sum_{i=1}^n X_{ik} = 1 \quad \forall k \tag{3}$$

  **for** $t \leftarrow T^{2/3}$ to $T - 1$ **do**
    At time $t$, use fractional allocation $X^t = \hat{X}$.
  **end for**

---

**Theorem 1.** *Suppose we are given $n, m, T, a, b$ such that $0 < a \leq \mu_{ik}^* \leq b$ for all $i \in [n]$, $k \in [m]$. If $\{\{(B_\ell(\mu), c_\ell)\}_{\ell=1}^L\}_\mu = \{\text{efe}(\mu)\}_\mu$ or $\{\{(B_\ell(\mu), c_\ell)\}_{\ell=1}^L\}_\mu = \{\text{pe}(\mu)\}_\mu$, then Algorithm 1 with probability $1 - 1/T$ satisfies the constraints $\{(B_\ell(\mu^*), c_\ell)\}_{\ell=1}^L$ and achieves regret of $\tilde{O}(T^{2/3})$ for constraints $\{(B_\ell(\mu^*), c_\ell)\}_{\ell=1}^L$.*

*Proof sketch.* We have already shown in Section 3 that both envy-freeness in expectation and proportionality in expectation satisfy Properties 1, 2, 3, and 4. Therefore, it suffices to show that Algorithm 1 achieves $\tilde{O}(T^{2/3})$ regret for any family of constraints $\{\{(B_\ell(\mu), c_\ell)\}_{\ell=1}^L\}_\mu$ that satisfies Properties 1, 2, 3, and 4.

The allocations used during the warm-up steps of Algorithm 1 are uniform at random, and therefore these allocations satisfy the constraints $\{(B_\ell(\mu), c_\ell)\}_{\ell=1}^L$ for all $\mu$ by Property 1. Because the fractional allocations used in the first $T^{2/3}$ steps are all UAR, each arm, or (player, item) pair, will be sampled with probability $\frac{1}{nm}$ at each step. This implies by Hoeffding's inequality that with high probability, $N_{ik} = \tilde{\Omega}(T^{2/3})$ for all $i, k$. The $i, k$ entry in the $\epsilon$ matrix is proportional to $\frac{1}{\sqrt{N_{ik}}}$, and

therefore $\|\epsilon\|_1 = \tilde{O}(T^{-1/3})$ with high probability. Because the value distributions are sub-Gaussian, a standard application of Hoeffding's inequality also gives that with high probability, the true mean matrix will be within our confidence region, i.e. $\mu^* \in \hat{\mu} \pm \epsilon$. To summarize, because we used UAR for the first $T^{2/3}$ steps, we have that

$$\Pr\left(\|\epsilon\|_1 \le \tilde{O}(T^{-1/3}), \mu^* \in \hat{\mu} \pm \epsilon\right) \ge 1 - \frac{1}{T}.$$

For the rest of the proof we will assume that the high probability event in the equation above holds. The next step is to show that $\hat{X}$ has $\|\epsilon\|_1$ per-step regret compared to $Y^{\mu^*}$. Let $K$ be the Lipschitz constant of Property 3. Using Property 2 with $\mu = \mu^*$ and $\gamma = 2K\|\epsilon\|_1 = \tilde{O}(T^{-1/3})$ gives that there exists an allocation $X'$ such that the social welfare of $X'$ is only $O(\|\epsilon\|_1)$ less than the social welfare of the optimal allocation $Y^{\mu^*}$ and such that every constraint $\ell$ either has slack of at least $2K\|\epsilon\|_1$ (option 1 of Property 2) or every player is treated equally in constraint $\ell$ (option 2 of Property 2). We will now show that $X'$ is a solution to LP (3). If option 1 holds for constraint $\ell \in [L]$, then by Property 3, $X'$ will satisfy the constraint $(B_\ell(\mu), c_\ell)$ for every $\mu \in \hat{\mu} \pm \epsilon$. Formally, if option 1 holds for constraint $\ell$, then for any $\mu \in \hat{\mu} \pm \epsilon$,

$$
\begin{aligned}
\langle B_\ell(\mu), X'\rangle_F &= \langle B_\ell(\mu), X'\rangle_F - \langle B_\ell(\mu^*), X'\rangle_F + \langle B_\ell(\mu^*), X'\rangle_F \\
&= \langle B_\ell(\mu) - B_\ell(\mu^*), X'\rangle_F + \langle B_\ell(\mu^*), X'\rangle_F \\
&\ge \langle B_\ell(\mu^*), X'\rangle_F - \|B_\ell(\mu) - B_\ell(\mu^*)\|_1 && [0 \le X'_{ik} \le 1, \ \forall i,k] \\
&\ge \langle B_\ell(\mu^*), X'\rangle_F - 2K\|\epsilon\|_1 && [\text{Property 3}] \\
&\ge c_\ell. && [\text{Property 2: option 1}]
\end{aligned}
$$

If option 2 holds for constraint $\ell \in [L]$, then Properties 1 and 4 together guarantee that $X'$ will satisfy the constraint $(B_\ell(\mu), c_\ell)$ for every $\mu \in \hat{\mu} \pm \epsilon$. Therefore, $X'$ will satisfy all of the constraints $\{B_\ell(\mu), c_\ell\}_{\ell=1}^L$ for every $\mu \in \hat{\mu} \pm \epsilon$, which implies that $X'$ is a solution to LP (3).

Finally, because $\hat{X}$ is the optimal solution to LP (3), $\hat{X}$ must have higher social welfare than $X'$ under means $\hat{\mu}$. Because $\|\mu^* - \hat{\mu}\|_1 \le \|\epsilon\|_1$, this implies that $\hat{X}$ must have at most $O(\|\epsilon\|_1)$ less social welfare than $X'$ under the true means $\mu^*$. An application of the triangle inequality gives that,

$$
\begin{aligned}
\langle Y^{\mu^*}, \mu^*\rangle_F - \langle \hat{X}, \mu^*\rangle_F &= \langle Y^{\mu^*}, \mu^*\rangle_F - \langle X', \mu^*\rangle_F + \langle X', \mu^*\rangle_F - \langle \hat{X}, \mu^*\rangle_F \\
&= \tilde{O}\left(\|\epsilon\|_1 + \|\epsilon\|_1\right) \\
&= \tilde{O}(T^{-1/3}).
\end{aligned}
$$

Thus, the total regret for the steps after the warm-up period is $\tilde{O}(T^{2/3})$. The regret of the warm-up period is at most $(b-a)T^{2/3}$ due to the assumption that the mean values are bounded. We can therefore conclude that the total regret is $\tilde{O}(T^{2/3})$, and this completes the proof of regret. Finally, we note that by construction of LP (3), if $\mu^* \in \hat{\mu} \pm \epsilon$ then the chosen fractional allocations $\hat{X}$ must also satisfy the constraints $\{(B_\ell(\mu^*), c_\ell)\}_{\ell=1}^L$ as desired. See Appendix D for the full proof. □

## 5 Lower bounds

The following lower bound shows that Theorem 1 is tight up to $\log$ factors. An equivalent result holds for proportionality, with the same proof.

**Theorem 2.** *There exists $a, b, n, m$ such that no algorithm can, for all $\mu^* \in [a, b]^{n \times m}$, both satisfy the envy-freeness constraints and achieve regret of less than $\frac{T^{2/3}}{\log(T)}$ with probability at least $1 - 1/T$. The same is true for the proportionality constraints.*

*Proof sketch.* We defer the formal proof to Appendix G and provide brief intuition here.

Suppose there are two item types and two players. In this case envy-freeness and proportionality are equivalent, and therefore we will focus on the former. Consider the following two mean value matrices.

$$\mu_1 = \begin{bmatrix} 2 & 3 \\ 1 & 1 \end{bmatrix} \qquad\qquad \mu_2 = \begin{bmatrix} 2 & 3 \\ 1 & 1 + T^{-1/3} \end{bmatrix}$$

We will show that no algorithm can with probability $1 - 1/T$ achieve regret of less than $\tilde{\Omega}(T^{2/3})$ and satisfy envy-freeness in expectation for both of these distributions. First, note that the expected social welfare maximizing allocation for $\mu_1$ is to give all items of type 1 to player 2 and all items of type 2 to player 1. On the other hand, any envy-free allocation for $\mu_2$ must give $\tilde{\Omega}(T^{-1/3})$ fraction of items of type 2 to player 2. This implies that if an algorithm is unable to distinguish between $\mu_1$ and $\mu_2$, then either the regret will be $\tilde{\Omega}(T^{2/3})$ for $\mu_1$ or the algorithm will not be envy-free for $\mu_2$.

Therefore, any algorithm that has regret of less than $\tilde{\Omega}(T^{2/3})$ and satisfies envy-freeness for both $\mu_1$ and $\mu_2$ must distinguish betwen $\mu_1$ and $\mu_2$. The only way to do this is to allocate at least $\tilde{\Omega}(T^{2/3})$ items of type 2 to player 2. However, this will result in regret under $\mu_1$ of $\tilde{\Omega}(T^{2/3})$. $\qquad\square$

## 6 Discussion

We conclude by discussing some limitations and open questions. First, Algorithm 1 involves solving a linear program with an infinite number of linear constraints. Linear programs with an infinite number of constraints (called semi-infinite programs) are well-studied and occur in many applications [16, 24]. We also note that a finite number of (exponentially many) constraints suffices for envy-freeness in expectation and proportionality in expectation by bounding all of the possible extreme values of $\epsilon$. Nevertheless, we opted to avoid this representation because it significantly complicates the presentation of the algorithm. Furthermore, there also exists a polynomial time separation oracle for determining whether an allocation satisfies the infinitely many constraints, which would allow techniques such as the Ellipsoid Method [6] to solve the linear program in polynomial time.

Second, while the regret coefficients for proportionality are polynomial in $n$, a practical limitation of Algorithm 1 for envy-freeness is that the $\tilde{O}$ term is exponential in $n$. We expect, however, that the worst-case bound we present in Lemma 1 is far from tight. Whether there exists a bound on the regret that is polynomial in $n$ for learning under envy-freeness in expectation constraints is an open question for future work.

The other natural question that remains open for future work is whether we can achieve $\tilde{O}(\sqrt{T})$ regret while maintaining envy-freeness in expectation or proportionality in expectation. If the optimal solution $Y^{\mu^*}$ has a positive slack in every constraint, then an upper confidence bound (UCB) approach would be likely to work. Unfortunately, the fairness constraints for envy-freeness in expectation and proportionality in expectation are often tight for the optimal allocation. Furthermore, the constraints have a constant (greater than $1/n$) dependence on every unknown value in the $\mu^*$ matrix. Therefore, every mean value $\mu_{ik}^*$ might need to be learned with high accuracy even if the optimal allocation does not allocate item type $k$ to player $i$.

We also note that there exist additional (albeit less prominent) fairness notions for the problem of online fair division, such as equitability, which may satisfy additional properties that allow for lower regret. We leave the question of studying more fairness notions for future work.

Finally, a broader question is whether the connection we have established between multi-armed bandits and online fair division might be leveraged to give a fresh perspective on additional problems in this area, such as online cake cutting [31].

## Acknowledgments and Disclosure of Funding

Procaccia gratefully acknowledges research support by the National Science Foundation under grants IIS-2147187, IIS-2229881 and CCF-2007080; and by the Office of Naval Research under grant N00014-20-1-2488. Schiffer was supported by an NSF Graduate Research Fellowship. Zhang was supported by an NSF Graduate Research Fellowship.

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

# A  Algorithmic Representation of Model

**Algorithm 2** [Online Item Allocation]

**Require:** ALG
1: $\forall i$, $A_i^0 \leftarrow \{\}$, $H_0 \leftarrow \{\}$
2: **for** $t \leftarrow 1$ to $T$ **do**
3:     $X_t \leftarrow \mathrm{ALG}(H_t)$
4:     $k_t \sim \mathcal{D}$
5:     Generate item $j_t$ of type $k_t$ (i.e. $V_i(j_t) \sim N(\mu_{ik_t}^*, 1)$, $\forall i \in N$ )
6:     $i_t \leftarrow$ Sample from $(X_t)_{k_t}^\top$
7:     $A_{i_t}^t = A_{i_t}^{t-1} + \{j_t\}$
8:     $H_t \leftarrow H_{t-1} + (k_t, i_t, V_{i_t}(j_t))$
9: **end for**
10: **return** $A = (A_1^T, A_2^T, ..., A_n^T)$

# B  Motivating Fairness in Expectation

In this section, we will explore the relationship between envy-freeness in expectation and realized envy, as well as the relationship between requiring envy-freeness in expectation at every time step and only requiring envy-freeness in expectation at the end of round $T$. For this section, we will use the following notation. For any algorithm ALG, we denote $X_t = \mathrm{ALG}(H_t)$ as the fractional allocation used at time $t$, and $\mathrm{Pr}_{\mathrm{ALG}}(i, k, H_t) := (X_t)_{ik}$.

Previous works on fair online allocation of indivisible goods have focused on the fairness of the final, realized allocation instead of studying fairness in expectation as in Definitions 1 and 2 [5]. We define the realized envy of an allocation below.

**Definition 4.** The realized envy of allocation $A$ at time $\tau$ is $\max_{i,i'} V_i(A_{i'}^\tau) - V_i(A_i^\tau)$.

We show in the following theorems that algorithms which are envy-free in expectation are within a $\log(T)$ factor of being "optimal" in terms of final realized envy. Informally, Theorem 3 shows that in our setting, no algorithm can with high probability output an allocation $A(\mathrm{ALG})$ with realized envy less than $\sqrt{t}$. Conversely, Theorem 4 shows that any algorithm ALG that satisfies envy-freeness in expectation will output a final allocation $A(\mathrm{ALG})$ with realized envy of at most $\sqrt{T}\log(T)$ with high probability.

**Theorem 3.** *Suppose $\mu^*$ is known. For any algorithm* ALG *and for any $\tau \in [T]$, with probability at least $1/16$ the allocation $A^\tau(\mathrm{ALG})$ has realized envy of more than $\sqrt{\tau}$.*

*proof.* Assume there are two players and only one item type, and assume that all values are drawn from $\mathcal{N}(\mu, 1)$. Fix $\tau \geq \log^2(T)$. As in Algorithm 2, define $i_t \in \{1, 2\}$ as the player allocated the item at time $t$. The realized envy of the two players can be written as:

$$\text{Realized Envy of Player 1 at time } \tau = \left( \sum_{t=0}^\tau \mathbb{1}_{i_t=1} \cdot V_1(j_t) \right) - \left( \sum_{t=0}^\tau \mathbb{1}_{i_t=2} \cdot V_1(j_t) \right) \quad (4)$$

$$\text{Realized Envy of Player 2 at time } \tau = \left( \sum_{t=0}^\tau \mathbb{1}_{i_t=2} \cdot V_2(j_t) \right) - \left( \sum_{t=0}^\tau \mathbb{1}_{i_t=1} \cdot V_2(j_t) \right). \quad (5)$$

Let $v_i^a$ be the value of the assigned player for item $i$, and $v_i^u$ be the value of the unassigned player for item $i$. Note that $v_i^a, v_i^u \sim N(\mu, 1)$, because at the time of allocation, ALG does not know either player's value for the item, and player values are therefore independent of the assignment.

By this coupling argument, Equations (4) and (5) can be rewritten as:

$$\text{Realized Envy of Player 1 at time } \tau = \left( \sum_{t=0}^\tau 1_{i_t=1} \cdot v_i^a \right) - \left( \sum_{i=0}^\tau 1_{i_t=2} \cdot v_i^u \right) \quad (6)$$

$$\text{Realized Envy of Player 2 at time } \tau = \left( \sum_{t=0}^{\tau} \mathbb{1}_{i_t=2} \cdot v_i^a \right) - \left( \sum_{t=0}^{\tau} \mathbb{1}_{i_t=1} \cdot v_i^u \right). \tag{7}$$

By the Central Limit Theorem, $\sum_{i=0}^{\tau} v_i^a$ and $\sum_{i=0}^{\tau} v_i^u$ are both $N(\tau \cdot \mu, \tau)$. Therefore, for any $\tau$, with probability at least $1/16$, we have that

$$\sum_{t=0}^{\tau} v_t^a \leq \tau\mu - \sqrt{\tau} \quad \text{and}$$

$$\sum_{t=0}^{\tau} v_t^u \geq \tau\mu + \sqrt{\tau}.$$

Putting these equations together, we have that

$$\sum_{t=0}^{\tau} v_t^a + 2\sqrt{\tau} \leq \sum_{t=0}^{\tau} v_t^u.$$

However, this result with Equations (6) and (7) implies that the envy must be at least $\sqrt{\tau}$ for any choice of $\{i_t\}$. Therefore, we have shown that with probability at least $1/16$, the envy at time $\tau$ will be at least $\sqrt{\tau}$ for any possible algorithm. $\square$

**Theorem 4.** *Suppose $\mu^*$ is known. Also assume that all of the value distributions are bounded by a constant $B$. If ALG is deterministic (i.e. $\mathrm{ALG}(H_t)$ is a deterministic function of $H_t$) and satisfies envy-freeness in expectation, then with probability $1 - o(1/T)$, the realized envy of $A^\tau(\mathrm{ALG})$ at every time $\tau \in [T]$ is at most $\sqrt{\tau}\log(T)$.*

*proof.* We will bound the realized envy of $A(\mathrm{ALG})$ between any two specific players $i, i'$, which will then imply a bound on the realized envy of $A(\mathrm{ALG})$ by a union bound. The key observation is that each round of Algorithm 2 consists of first a random draw from $\mathcal{D}$ to determine the item type $k_t$ and then a draw from $\xi_t \sim \mathrm{Unif}([0,1])$ which determines the player to which the item is allocated based on $X_t = \mathrm{ALG}(H_t)$. Formally, the $t^{\mathrm{th}}$ item is allocated to player $i$ if

$$\xi_t \in \left[ \sum_{i''=1}^{i-1} (X_t)_{i''k_t} \;,\; \sum_{i''=1}^{i} (X_t)_{i''k_t} \right].$$

Define $\{v_{ti''}\}_{i''=1}^n$ as the values of the players for the item at time $t$. This is the final source of randomness in round $t$. Therefore, the allocation of any player at time $\tau$ is a function of the random variable sequence $\{(k_t, \xi_t, \{v_{ti''}\}_{i''=1}^n)\}_{t=0}^{\tau-1}$.

Let $E_\tau$ represent the envy accrued by player $i$ for player $i'$ up until but not including time $\tau$. Then

$$E_\tau = \sum_{t=0}^{\tau-1} v_{ti} \cdot \sum_{k=1}^{m} \mathbb{1}_{k_t=k} \cdot \left( \mathbb{1}_{\xi_t \in [\sum_{s=1}^{i'-1}(X_t)_{sk_t}, \sum_{s=1}^{i'}(X_t)_{sk_t}]} - \mathbb{1}_{\xi_t \in [\sum_{s=1}^{i-1}(X_t)_{sk_t}, \sum_{s=1}^{i}(X_t)_{sk_t}]} \right)$$
$$:= f\left( \{(k_t, \xi_t, \{v_{ti''}\}_{i''=1}^n)\}_{t=0}^{\tau-1} \right).$$

Now we will apply McDiarmid's inequality to the function $f\left( \{(k_t, \xi_t, \{v_{ti''}\}_{i''=1}^n)\}_{t=0}^{\tau-1} \right)$. First, we show that the bounded condition is satisfied. If $\{(k_t, \xi_t, \{v_{ti''}\}_{i''=1}^n)\}_{t=0}^{\tau-1}$ and $\{(k_t', \xi_t', \{v_{ti''}'\}_{i''=1}^n)\}_{t=0}^{\tau-1}$ differ only at the $s^{\mathrm{th}}$ element for any $s \in [0, \tau-1]$, then

$$\left| f\left( \{(k_t, \xi_t, \{v_{ti''}\}_{i''=1}^n)\}_{t=0}^{\tau-1} \right) - f\left( \{(k_t', \xi_t', \{v_{ti''}'\}_{i''=1}^n)\}_{t=0}^{\tau-1} \right) \right| \leq B.$$

Therefore, we can apply McDiarmid's inequality to get that

$$\Pr\left( \left| f\left( \{(k_t, \xi_t, \{v_{ti''}\}_{i''=1}^n)\}_{t=0}^{\tau-1} \right) - \mathbb{E}\left[ f\left( \{(k_t, \xi_t, \{v_{ti''}\}_{i''=1}^n)\}_{t=0}^{\tau-1} \right) \right] \right| \geq \sqrt{\tau}\log(T) \right) \leq e^{-\log^2(T)/(2B^2)}$$
$$\leq T^{-\log(T)/(2B^2)}.$$

Since ALG is envy-free in expectation, we know that $\mathbb{E}\left[f\left(\{(k_t, \xi_t, \{v_{ti''}\}_{i''=1}^n)\}_{t=0}^{\tau-1}\right)\right] \leq 0$. Therefore, we must have that

$$\Pr\left(f\left(\{(k_t, \xi_t, \{v_{ti''}\}_{i''=1}^n)\}_{t=0}^{\tau-1}\right) \geq \sqrt{\tau}\log(T)\right) \leq T^{-\log(T)/(2B^2)}.$$

Taking a union bound over all pairs $i, i'$ and all $\tau \in [T]$, we have that

$$\Pr(\exists \tau : \text{Realized envy at time } \tau \text{ is} \geq \sqrt{\tau}\log(T)) \leq n^2 T \cdot T^{-2\log(T)/b^2} = o(1/T).$$

Therefore, with probability $1 - o(1/T)$, the realized envy at evert time $\tau$ is at most $\sqrt{\tau}\log(T)$. $\quad\square$

Note that Theorem 3 does not contradict the results of Benadè et al. [5], who achieve envy-freeness of the realized allocation with high probability when the true player values for the items are known at time of allocation (as opposed to our model which only knows the item type). When player values for item $j_t$ are known before assignment, Theorem 3 does not apply.

We also defined envy-freeness in expectation as a constraint which needs to hold at every time step $t$. In some fair division applications, we may only care about "fairness" of the total allocation at the end of the process. Therefore, we could instead only require that no player is envious in expectation of any other player at the end of all $T$ rounds. However, Theorem 5 shows that this is equivalent to maintaining envy-freeness in expectation at all times $t \in [T]$ when maximizing social welfare.

**Theorem 5.** *Suppose $\mu^*$ is known. Let $\mathcal{E}$ be the class of all algorithms that are envy-free in expectation and let $\mathcal{F}$ be the class of all algorithms which satisfy $\mathbb{E}[V_i(A_i^T)] \geq \max_{i' \in [n]} \mathbb{E}[V_i(A_{i'}^T)]$ for all $i$. Then*

$$\max_{\text{ALG} \in \mathcal{F}} \mathbb{E}[\text{sw}(A(\text{ALG}))] = \max_{\text{ALG} \in \mathcal{E}} \mathbb{E}[\text{sw}(A(\text{ALG}))].$$

*proof.* By definition, $\mathcal{E} \subseteq \mathcal{F}$, which proves one direction of the desired equality. We will now show that for any $\text{ALG} \in \mathcal{F}$, there exists an algorithm $\text{ALG}' \in \mathcal{E}$ such that

$$\mathbb{E}[\text{sw}(A(\text{ALG}'))] = \mathbb{E}[\text{sw}(A(\text{ALG}))].$$

First, observe that by the definition of $\mathcal{F}$, we have that for all $i, i'$,

$$\frac{1}{T}\mathbb{E}\left[\sum_{t \in [T]} \sum_k \Pr_{\text{ALG}}(i, k, H_t)\mu_{ik}\Pr_{\mathcal{D}}(k)\right] \geq \frac{1}{T}\mathbb{E}\left[\sum_{t \in [T]} \sum_k \Pr_{\text{ALG}}(i', k, H_t)\mu_{ik}\Pr_{\mathcal{D}}(k)\right]. \quad (8)$$

By linearity of expectation, Equation (8) is equivalent to

$$\frac{1}{T}\sum_{t \in [T]} \sum_k \int_{H_t} \Pr_{\text{ALG}}(i, k, H_t)dH_t \cdot \mu_{ik}\Pr_{\mathcal{D}}(k) \geq \frac{1}{T}\sum_{t \in [T]} \sum_k \int_{H_t} \Pr_{\text{ALG}}(i', k, H_t)dH_t \cdot \mu_{ik}\Pr_{\mathcal{D}}(k).$$
$$(9)$$

Furthermore, by definition

$$\mathbb{E}[\text{sw}(\text{ALG})] = \mathbb{E}\left[\sum_{i=1}^n \sum_{t \in [T]} \sum_k \Pr_{\text{ALG}}(i, k, H_t)\mu_{ik}\Pr_{\mathcal{D}}(k)\right]$$
$$= \sum_{i=1}^n \sum_{t \in [T]} \sum_k \int_{H_t} \Pr_{\text{ALG}}(i, k, H_t)dH_t \cdot \mu_{ik} \cdot \Pr_{\mathcal{D}}(k)$$
$$= \sum_{i=1}^n \sum_k \left(\sum_{t \in [T]} \int_{H_t} \Pr_{\text{ALG}}(i, k, H_t)dH_t\right) \cdot \mu_{ik} \cdot \Pr_{\mathcal{D}}(k).$$

We will construct the algorithm $\text{ALG}'$ as follows. Suppose $\text{ALG}'$ is time-independent and history-independent, such that for all $t, H_t$,

$$\Pr_{\text{ALG}'_t}(i, k, H_t) = \frac{1}{T}\sum_{s \in [T]} \int \Pr_{\text{ALG}_s}(i, k, H_s)dH_s.$$

The expected social welfare of ALG′ is

$$\mathbb{E}[\mathrm{sw}(\mathrm{ALG}')] = \mathbb{E}\left[\sum_{i=1}^{n}\sum_{t\in[T]}\sum_{k}\Pr_{\mathrm{ALG}'_t}(i,k,H_t)\mu_{ik}\Pr_{\mathcal{D}}(k)\right]$$

$$= \mathbb{E}\left[\sum_{i=1}^{n}\sum_{t\in[T]}\sum_{k}\frac{1}{T}\sum_{s\in[T]}\int\Pr_{\mathrm{ALG}_s}(i,k,H_s)dH_s\cdot\mu_{ik}\cdot\Pr_{\mathcal{D}}(k)\right]$$

$$= \mathbb{E}\left[\sum_{i=1}^{n}\sum_{k}\left(\sum_{s\in[T]}\int\Pr_{\mathrm{ALG}_s}(i,k,H_s)dH_s\right)\cdot\mu_{ik}\cdot\Pr_{\mathcal{D}}(k)\right]$$

$$= \mathbb{E}[\mathrm{sw}(\mathrm{ALG})].$$

Therefore, ALG and ALG′ have the same expected social welfare. Finally, we need to show that ALG′ $\in \mathcal{E}$. This is equivalent to showing that for all $t$ and $H_t$,

$$\sum_{k}\Pr_{\mathrm{ALG}'_t}(i,k,H_t)\mu_{ik}\Pr_{\mathcal{D}}(k) \geq \sum_{k}\Pr_{\mathrm{ALG}'_t}(i',k,H_t)\mu_{ik}\Pr_{\mathcal{D}}(k).$$

Starting with the LHS and plugging in the definition of ALG′, we have that

$$\sum_{k}\Pr_{\mathrm{ALG}'_t}(i,k,H_t)\mu_{ik}\Pr_{\mathcal{D}}(k) = \sum_{k}\frac{1}{T}\sum_{s\in[T]}\int\Pr_{\mathrm{ALG}_s}(i,k,H_s)dH_s\mu_{ik}\Pr_{\mathcal{D}}(k)$$

$$= \frac{1}{T}\sum_{k}\sum_{s\in[T]}\int\Pr_{\mathrm{ALG}_s}(i,k,H_s)dH_s\mu_{ik}\Pr_{\mathcal{D}}(k)$$

$$\geq \frac{1}{T}\sum_{k}\sum_{s\in[T]}\int\Pr_{\mathrm{ALG}_s}(i',k,H_s)dH_s\mu_{ik}\Pr_{\mathcal{D}}(k) \quad [\text{Equation (9)}]$$

$$= \sum_{k}\frac{1}{T}\sum_{s\in[T]}\int\Pr_{\mathrm{ALG}_s}(i',k,H_s)dH_s\mu_{ik}\Pr_{\mathcal{D}}(k)$$

$$= \sum_{k}\Pr_{\mathrm{ALG}'_t}(i',k,H_t)\mu_{ik}\Pr_{\mathcal{D}}(k),$$

as desired. Therefore, we have shown that ALG′ $\in \mathcal{E}$. $\qquad\square$

**Theorem 6.** *The algorithm which maximizes expected social welfare subject to* EFE *up to time $T$ is time-independent, history-independent, and can be calculated in polynomial time.*

*proof.* By Theorem 5, there exists an optimal algorithm that satisfies envy-freeness in expectation and that is time-independent and history-independent. To find the best such fractional allocation, all we must do is solve LP (1) with the envy-freeness in expectation constraints. $\qquad\square$

### B.1 Proportionality

Theorems 3–6 also have equivalent forms for proportionality. We define the realized proportionality gap as the equivalent of envy for the proportionality constraints. This implies that a proportional allocation has non-positive proportionality gap.

**Definition 5.** The realized proportionality gap of allocation $A$ at time $\tau$ is $\max_i \frac{1}{n}\sum_{i'}V_i(A_{i'}^{\tau}) - V_i(A_i^{\tau})$.

As in Theorems 3 and 4 , the following two results give that algorithms which satisfy proportionality in expectation are within a $\log(T)$ factor of optimal for the realized proportionality gap.

**Theorem 7.** *Suppose $\mu^*$ is known. For any algorithm* ALG *and for any $\tau \in [T]$, with probability at least $1/16$ the allocation $A^{\tau}(\mathrm{ALG})$ has realized proportionality gap of more than $\sqrt{\tau}$.*

*proof.* As in the proof of Theorem 3, assume there are two players and only one item type, and assume that all values are drawn from $\mathcal{N}(\mu, 1)$. Then the realized proportionality gap of the two players can be written as:

$$\text{Realized proportionality gap of Player 1 at time } \tau = \frac{1}{2}\left(\sum_{t=0}^{\tau} \mathbb{1}_{i_t=1} \cdot V_1(j_t)\right) - \frac{1}{2}\left(\sum_{t=0}^{\tau} \mathbb{1}_{i_t=2} \cdot V_1(j_t)\right) \tag{10}$$

$$\text{Realized proportionality gap of Player 2 at time } \tau = \frac{1}{2}\left(\sum_{t=0}^{\tau} \mathbb{1}_{i_t=2} \cdot V_2(j_t)\right) - \frac{1}{2}\left(\sum_{t=0}^{\tau} \mathbb{1}_{i_t=1} \cdot V_2(j_t)\right) \tag{11}$$

These equations only differ from those of realized envy by a scalar factor, and therefore the rest of the proof follows exactly as in Theorem 3. $\qquad \square$

**Theorem 8.** *Suppose $\mu^*$ is known. Also assume that all of the value distributions are bounded by a constant $B$. If $\mathrm{ALG}$ is deterministic (i.e. $\mathrm{ALG}(H_t)$ is a deterministic function of $H_t$) and satisfies proportionality in expectation, then with probability $1 - o(1/T)$, the realized proportionality gap of $A^\tau(\mathrm{ALG})$ at every time $\tau \in [T]$ is at most $\sqrt{\tau}\log(T)$.*

*proof.* In this proof, we can let $E_t$ be the accrued "proportionality gap" of any player $i$. Then as in Theorem 4, an application of McDiarmid's inequality allows us to bound the realized proportionality gap with high probability for all $\tau$. $\qquad \square$

The following two theorems are analogs of Theorem 5 and Theorem 6 for envy-freeness in expectation. Together, these theorems imply that maximizing social welfare subject to proportionality in expectation at every time step is equivalent to maximizing social welfare subject to proportionality in expectation only at the end of round $T$.

**Theorem 9.** *Suppose $\mu^*$ is known. Let $\mathcal{E}$ be the class of all algorithms that are proportional in expectation and let $\mathcal{F}$ be the class of all algorithms which satisfy $\mathbb{E}[V_i(A_i^T)] \geq \frac{1}{n}\sum_{i' \in [n]} \mathbb{E}[V_i(A_{i'}^T)]$ for all $i$. Then*

$$\max_{\mathrm{ALG} \in \mathcal{F}} \mathbb{E}[\mathrm{sw}(A(\mathrm{ALG}))] = \max_{\mathrm{ALG} \in \mathcal{E}} \mathbb{E}[\mathrm{sw}(A(\mathrm{ALG}))].$$

*proof.* Suppose $\mathrm{ALG} \in \mathcal{F}$. We will define $\mathrm{ALG}'$ as in Theorem 5 to be

$$\Pr_{\mathrm{ALG}'_t}(i, k, H_t) = \frac{1}{T}\sum_{s \in [T]} \int \Pr_{\mathrm{ALG}_s}(i, k, H_s)dH_s.$$

By this construction, $\mathrm{ALG}' \in \mathcal{E}$ because

$$\sum_k \Pr_{\mathrm{ALG}'_t}(i, k, H_t)\mu_{ik}\Pr_{\mathcal{D}}(k)$$

$$= \sum_k \frac{1}{T}\sum_{s \in [T]} \int \Pr_{\mathrm{ALG}_s}(i, k, H_s)dH_s\mu_{ik}\Pr_{\mathcal{D}}(k)$$

$$= \frac{1}{T}\sum_k \sum_{s \in [T]} \int \Pr_{\mathrm{ALG}_s}(i, k, H_s)dH_s\mu_{ik}\Pr_{\mathcal{D}}(k)$$

$$\geq \frac{1}{Tn}\sum_{i'=1}^{n}\sum_k \sum_{s \in [T]} \int \Pr_{\mathrm{ALG}_s}(i', k, H_s)dH_s\mu_{ik}\Pr_{\mathcal{D}}(k) \qquad [\mathrm{ALG} \in \mathcal{F}]$$

$$= \sum_{i'=1}^{n}\sum_k \frac{1}{Tn}\sum_{s \in [T]} \int \Pr_{\mathrm{ALG}_s}(i', k, H_s)dH_s\mu_{ik}\Pr_{\mathcal{D}}(k)$$

$$= \frac{1}{n}\sum_{i'=1}^{n}\sum_k \Pr_{\mathrm{ALG}'_t}(i', k, H_t)\mu_{ik}\Pr_{\mathcal{D}}(k).$$

Because $\mathcal{E} \subseteq \mathcal{F}$ and because we showed in Theorem 5 that ALG$'$ and ALG have the same expected social welfare, the desired result follows. $\qquad\square$

**Theorem 10.** *The algorithm which maximizes expected social welfare subject to* PE *up to time $T$ is time-independent, history-independent, and can be calculated in polynomial time.*

*proof.* By Theorem 9, there exists an optimal algorithm that satisfies proportionality in expectation that is time-independent and history-independent. To find the best such fractional allocation, all we must do is solve LP (1) with the proportionality in expectation constraints. $\qquad\square$

## C  Additional Model Notes

### C.1  Choice of $\mathcal{D}$

In the body of the paper, we focus on the case when $\mathcal{D}$ is the uniform distribution over item types. However, our results generalize to any distribution $\mathcal{D}$ which does not depend on $T$. In this case, the social welfare of a fractional allocation $X$ becomes $\sum_{i,k} X_{ik} \Pr_{\mathcal{D}}(k) \mu_{ik}$. For a matrix $\nu \in \mathbb{R}^{n \times m}$, define $f_{\mathcal{D}}(\nu) = \nu' \in \mathbb{R}^{n \times m}$, where $\nu'_{ik} = n \cdot \Pr_{\mathcal{D}}(k) \cdot \mu_{ik}$. For mean values $\mu^*$, define $\mu' = f_{\mathcal{D}}(\mu^*)$. Then social welfare of a fractional allocation $X$ with means $\mu^*$ and distribution $\mathcal{D}$ is then simply $\frac{1}{n}\langle X, \mu' \rangle_F$ as in the uniform $\mathcal{D}$ case. Similarly, the envy-freeness in expectation or proportionality in expectation constraints on $X$ with means $\mu^*$ and item distribution $\mathcal{D}$ can be represented as $\langle B_\ell(\mu'), X \rangle_F \geq c_\ell$, which is an equivalent form to the constraints when $\mathcal{D}$ is uniform. Therefore, for arbitrary $\mathcal{D}$ we can use Algorithm 1 with only two slight modifications. The first is we must transform the $\hat{\mu}, \mu$, and other components of the linear programs using the function $f_{\mathcal{D}}$. The second modification is that we potentially need more exploration steps (larger $T$) in the warm-up period to guarantee the same level of estimation of $\mu^*$, depending on the value of $\min_{i,k} \Pr_{\mathcal{D}}(k)$. However, since we require that $\mathcal{D}$ does not depend on $T$, this will not change the overall regret of the algorithm.

### C.2  Lower Bound on Means

In this section, we show that if the means of player values can be arbitrarily close to zero, then it can be impossible to achieve a regret of $o(T)$.

**Theorem 11.** *For $\epsilon > 0$, there does not exist an algorithm* ALG *such that for any possible $\mu^* \in [0, \epsilon]^{n \times m}$, for every $t$ the fractional allocation $X_t$ chosen by* ALG *satisfies envy-freeness in expectation with probability greater than $1/2$ and the regret of* ALG *is $o(T)$. The same result also holds for proportionality in expectation.*

*proof.* W.l.o.g. assume that $\epsilon = 1$. The proof also holds for any other constant $\epsilon$. Suppose the underlying value distributions are Bernoulli, that we have two players and two item types, and assume $T \geq 2$. We will consider two cases for $\mu^*$ and show that no algorithm can with probability greater than $1/2$ satisfy the constraints and have regret of $o(T)$ for both of these cases of $\mu^*$.

First, let

$$\mu^1 = \begin{bmatrix} 1/T^2 & 0 \\ 1 & 0.5 \end{bmatrix} \quad \text{and}$$

$$\mu^2 = \begin{bmatrix} 0 & 1/T^2 \\ 1 & 0.5 \end{bmatrix}.$$

If $\mu^* = \mu^1$ or $\mu^* = \mu^2$, then player 1 will not have a realized value of 1 for any of the $T$ items with probability at least $1/2$. Therefore, no algorithm can differentiate between $\mu^* = \mu^1$ and $\mu^* = \mu^2$ with probability at least $1/2$. The only fractional allocation that is envy-free for both $\mu^* = \mu^1$ and $\mu^* = \mu^2$ is the uniform at random allocation. However, this allocation has regret of $\Omega(T)$ for $\mu^* = \mu^2$, as the best fractional allocation when $\mu^* = \mu^2$ is the fractional allocation

$$Y^{\mu^2} = \begin{bmatrix} 0 & 0.5 \\ 1 & 0.5 \end{bmatrix}.$$

This proves the desired result that no algorithm can be envy-free for $\mu^*$ and have $o(T)$ regret for both possible realizations of $\mu^*$. Similarly, the uniform at random allocation is the only proportional allocation in this example, and therefore the same result holds. $\qquad\square$

# D Proof of Theorem 1

Observations 1, 2, and 3 give that the proportionality in expectation constraints and the envy-freeness in expectation constraints satisfy Properties 1, 3, and 4 respectively. Lemmas 1 and 2 respectively give that the proportionality in expectation constraints and the envy-freeness in expectation constraints satisfy Property 2. These results combined with Lemma 3 directly prove Theorem 1.

**Lemma 3.** *Let $\left\{\{(B_\ell(\mu), c_\ell)\}_{\ell=1}^L\right\}_{\mu \in [a,b]^{n \times m}}$ be a family of constraints that satisfy Properties 1, 3, 4, and 2. Then with probability $1 - 1/T$, Algorithm 1 satisfies constraints $\{(B_\ell(\mu^*), c_\ell)\}_{\ell=1}^L$ and has regret of $\tilde{O}(T^{2/3})$ for constraints $\{(B_\ell(\mu^*), c_\ell)\}_{\ell=1}^L$.*

*proof.* First, we note that the regret of the first $T^{2/3}$ steps can be bounded by

$$\sum_{t=0}^{T^{2/3}-1} \langle Y^{\mu^*}, \mu^* \rangle_F - \langle X_t, \mu^* \rangle_F \leq T^{2/3}(b-a) = O(T^{2/3}). \tag{12}$$

By Property 1, the uniform at random allocation satisfies constraints $\{(B_\ell(\mu), c_\ell\}_{\ell=1}^L$. Therefore, $X_t$ satisfies the constraints for all $t < T^{2/3}$. Furthermore, because the fractional allocation was uniform at random for the first $T^{2/3}$ steps, we have that for sufficiently large $T$,

$$\Pr\left( \|\epsilon\|_1 \leq nm\sqrt{nm\log^2(4nmT)} \cdot T^{-1/3} \right)$$

$$\geq \Pr\left( \forall i \in [n], k \in [m], \epsilon_{ik} \leq \sqrt{nm\log^2(4nmT)} \cdot T^{-1/3} \right)$$

$$= \Pr\left( \forall i \in [n], k \in [m], N_{ik} \geq \frac{T^{2/3}}{2nm} \right)$$

$$= \Pr\left( \forall i \in [n], k \in [m], N_{ik} \geq \frac{T^{2/3}}{nm} - \frac{T^{2/3}}{2nm} \right)$$

$$\geq \Pr\left( \forall i \in [n], k \in [m], N_{ik} \geq \frac{T^{2/3}}{nm} - \sqrt{\log(4nmT)} \cdot T^{1/3} \right)$$

$$\geq 1 - nme^{-2\log(4nmT)}$$

$$\geq 1 - \frac{1}{2T}, \tag{13}$$

where the second to last inequality is by Hoeffding's Inequality and a union bound. This implies that with probability $1 - \frac{1}{2T}$, $\|\epsilon\|_1 = \tilde{O}(T^{-1/3})$. Because the values are drawn from a Sub-Gaussian distribution, there exists a constant $c > 0$ (which depends on the distribution of the values) such that by Hoeffding's inequality,

$$\Pr\left( \forall i \in [n], k \in [m], |\hat{\mu}_{ik} - \mu_{ik}^*| \leq \epsilon_{ik} \right) \geq 1 - 2nme^{-c\log^2(4nmT)}$$

$$\geq 1 - 2nm\left(\frac{1}{4nmT}\right)^{c\log(4nmT)}$$

$$\geq 1 - \frac{1}{2T}. \qquad \text{[For sufficiently large } T]$$

$$\tag{14}$$

For the rest of this proof, we will assume that

$$\|\epsilon\|_1 \leq \tilde{O}(T^{-1/3}) \tag{15}$$

and

$$\forall i \in [n], k \in [m], |\hat{\mu}_{ik} - \mu_{ik}^*| \leq \epsilon_{ik}, \tag{16}$$

which by Equations (13) and (14) happens with probability $1 - 1/T$.

If $K$ is the Lipschitz constant for this family of constraints, then by Equation (15), $2K\|\epsilon\|_1 \leq \tilde{O}(T^{-1/3}) \leq \gamma_0$ for sufficiently large $T$, where $\gamma_0$ is from Property 2. Therefore, taking $\gamma = 2K\|\epsilon\|_1$ in Property 2 gives that there exists some fractional allocation $X'$ such that

$$|\langle \mu^*, Y^{\mu^*} \rangle_F - \langle \mu^*, X' \rangle_F| \leq O(\|\epsilon\|_1), \tag{17}$$

and such that for every constraint $\ell \in [L]$, either $\forall i_1, i_2 \in \{i : B_\ell(\mu^*)_i \neq \mathbf{0}\}$, $X'_{i_1} = X'_{i_2}$ or

$$\langle B_\ell(\mu^*), X' \rangle_F \geq c_\ell + 2K\|\epsilon\|_1. \tag{18}$$

For any $\mu \in \hat{\mu} \pm \epsilon$, we have that $\|\mu - \mu^*\|_1 \leq 2\|\epsilon\|_1$ by Equation (16) and the triangle inequality. By the Lipschitz continuity assumption (Property 3), this implies that for all $\mu \in \hat{\mu} \pm \epsilon$,

$$\|B_\ell(\mu) - B_\ell(\mu^*)\|_1 \leq 2K\|\epsilon\|_1. \tag{19}$$

Therefore, if Equation (18) holds for a constraint $\ell$, then for any $\mu \in \hat{\mu} \pm \epsilon$,

$$
\begin{aligned}
\langle B_\ell(\mu), X' \rangle_F &= \langle B_\ell(\mu), X' \rangle_F - \langle B_\ell(\mu^*), X' \rangle_F + \langle B_\ell(\mu^*), X' \rangle_F \\
&= \langle B_\ell(\mu) - B_\ell(\mu^*), X' \rangle_F + \langle B_\ell(\mu^*), X' \rangle_F \\
&\geq \langle B_\ell(\mu^*), X' \rangle_F - \|B_\ell(\mu) - B_\ell(\mu^*)\|_1 \qquad &[0 \leq X'_{ik} \leq 1, \ \forall i, k] \\
&\geq \langle B_\ell(\mu^*), X' \rangle_F - 2K\|\epsilon\|_1 \qquad &[\text{Equation (19)}] \\
&\geq c_\ell. \qquad &[\text{Equation (18)}]
\end{aligned}
$$

Therefore, we have shown that if Equation (18) holds for a constraint $\ell$, then the fractional allocation $X'$ satisfies constraint $(B_\ell(\mu), c_\ell)$ for all $\mu \in \hat{\mu} \pm \epsilon$. If Equation (18) does not hold for a constraint $\ell$, then $\forall i_1, i_2 \in \{i : B_\ell(\mu^*)_i \neq \mathbf{0}\}$, $X'_{i_1} = X'_{i_2}$. Because $\{i : B_\ell(\mu^*)_i \neq \mathbf{0}\} = \{i : B_\ell(\mu)_i \neq \mathbf{0}\}$ by Property 4, this implies by Property 1 that $\langle B_\ell(\mu), X' \rangle_F \geq c_\ell$ for all $\mu$. Therefore, we have shown that $X'$ satisfies $\{(B_\ell(\mu), c_\ell)\}_{\ell=1}^L$ for all $\mu \in \hat{\mu} \pm \epsilon$, and thus $X'$ satisfies the constraints in LP (3).

Because $\hat{X}$ is the optimal solution to LP (3), we have that

$$\langle X', \hat{\mu} \rangle_F \leq \langle \hat{X}, \hat{\mu} \rangle_F.$$

By Equation (16), this implies that

$$\langle X', \mu^* \rangle_F - \langle \hat{X}, \mu^* \rangle_F \leq \|\epsilon\|_1. \tag{20}$$

Therefore,

$$
\begin{aligned}
\langle Y^{\mu^*}, \mu^* \rangle_F - \langle \hat{X}, \mu^* \rangle_F &= \langle Y^{\mu^*}, \mu^* \rangle_F - \langle X', \mu^* \rangle_F + \langle X', \mu^* \rangle_F - \langle \hat{X}, \mu^* \rangle_F \\
&\leq O\left(\|\epsilon\|_1 + \|\epsilon\|_1\right) \qquad &[\text{Equations (17) and (20)}] \\
&\leq \tilde{O}(T^{-1/3}). \qquad &[\text{Equation (15)}]
\end{aligned}
\tag{21}
$$

Combining with Equation (12), this gives a total regret of

$$
\begin{aligned}
\sum_{t=0}^{T-1} \left( \langle Y^{\mu^*}, \mu^* \rangle_F - \langle X^t, \mu^* \rangle_F \right) &\leq O(T^{2/3}) + \sum_{t=T^{2/3}}^{T} \left( \langle Y^{\mu^*}, \mu^* \rangle_F - \langle X^t, \mu^* \rangle_F \right) \qquad &[\text{Equation (12)}] \\
&= O(T^{2/3}) + T \cdot \tilde{O}(T^{-1/3}) \qquad &[\text{Equation (21)}] \\
&= \tilde{O}(T^{2/3}).
\end{aligned}
$$

Lastly, we must show that the constraints are satisfied by the fractional allocation used by the algorithm for $t \geq T^{2/3}$. This is because if Equation (16) holds, then any solution to LP (3) must satisfy the constraints $\{(B_\ell(\mu^*), c_\ell)\}_{\ell=1}^L$, and therefore the fractional allocation used by the algorithm for all $t \geq T^{2/3}$ will satisfy these constraints. Recall that all of the above relies on Equations (16) and (15) holding, which happens with probability $1 - 1/T$ as desired. $\qquad\square$

# E   Proof of Lemma 2

For proportionality in expectation, LP (1) can be rewritten as the following linear program.

$$
\begin{aligned}
Y^\mu := \arg\max \ & \langle X, \mu \rangle_F \\
\text{s.t. } & X_i \cdot \mu_i - \frac{1}{n}\|\mu_i\|_1 \geq 0 \quad \forall i \in [n] \\
& \sum_i X_{ik} = 1 \quad \forall k
\end{aligned}
\tag{22}
$$

In order to show that the proportionality constraints satisfy Property 2, we want to construct an $X'$ such that

1. $X'$ decreases the social welfare relative to $Y^\mu$ by $O(\gamma)$ and

2. For every constraint $i \in [n]$, either $X' = \mathrm{UAR}$ or

$$X'_i \cdot \mu_i - \frac{1}{n}\|\mu_i\|_1 \geq \gamma. \tag{23}$$

LP (22) has $n$ constraints, one corresponding to each player. Define

$$S_i = Y^\mu_i \cdot \mu_i - \frac{1}{n}\|\mu_i\|_1. \tag{24}$$

$S_i$ is the slack on the $i$th constraint when using the optimal solution $Y^\mu$. Now we have two cases depending on $\sum_{i=1}^n S_i$, the total amount of slack across all $n$ players.

**Case 1:** $\sum_{i=1}^n S_i \leq \frac{b}{a} n\gamma$

Let $X' = \mathrm{UAR}$. This will result in an decrease of social welfare of at most $\frac{b}{a} n\gamma$ compared to $Y^\mu$. To see why, note that the slack of constraint $i$ is equivalent to how much player $i$ prefers their fractional allocation in $Y^\mu$ to UAR. Therefore, switching to UAR from $Y^\mu$ decreases the total social welfare by $S_i$ per player, and therefore decreases the total social welfare by $\sum_{i=1}^n S_i \leq \frac{b}{a} n\gamma = O(\gamma)$. Furthermore, $X' = \mathrm{UAR}$ clearly satisfies the other condition because every player is treated equally.

**Case 2:** $\sum_{i=1}^n S_i > \frac{b}{a} n\gamma$

Intuitively, in this case we want to redistribute the slack from the constraints with a lot of slack to the constraints without much slack. To do this, construct $X'$ as follows. Define

$$\Delta_{ik} := \frac{Y^\mu_{ik}}{\sum_{k'=1}^m Y^\mu_{ik'}} \cdot \frac{S_i}{\sum_{i'=1}^n S_{i'}} \cdot \frac{n\gamma}{a}. \tag{25}$$

By construction, we have that

$$\sum_{i=1}^n \sum_{k=1}^m \Delta_{ik} = \frac{n\gamma}{a}. \tag{26}$$

Because $\sum_{i=1}^n S_i \geq \frac{b}{a} n\gamma$, we also have that

$$\frac{S_i}{\sum_{i'} S_{i'}} \cdot \frac{n\gamma}{a} \leq \frac{S_i}{b}. \tag{27}$$

Furthermore, for every $i$, $S_i \leq Y^\mu_i \cdot \mu_i$ by definition of $S_i$. Because $\mu_{ik} \leq b$, this implies that $\frac{S_i}{b} \leq \sum_{k=1}^m Y^\mu_{ik}$. With Equation (27), this implies that $\frac{S_i}{\sum_{i'} S_{i'}} \cdot \frac{n\gamma}{a} \leq \sum_{k=1}^m Y^\mu_{ik}$. With Equation (25), this implies that $\Delta_{ik} \leq Y^\mu_{ik}$. Finally, we note that

$$
\begin{aligned}
\Delta_i \cdot \mu_i &= \frac{Y^\mu_i \cdot \mu_i}{\sum_{k'=1}^m Y^\mu_{ik'}} \cdot \frac{S_i}{\sum_{i'=1}^n S_{i'}} \cdot \frac{n\gamma}{a} && \text{[Equation (25)]} \\
&\leq \left(\frac{\sum_{k=1}^m Y^\mu_{ik}\mu_{ik}}{b\sum_{k'=1}^m Y^\mu_{ik'}}\right) S_i && \text{[Equation (27)]} \\
&\leq \left(\frac{\sum_{k=1}^m Y^\mu_{ik}}{\sum_{k'=1}^m Y^\mu_{ik'}}\right) S_i && [\mu_{ik} \leq b] \\
&= S_i. 
\end{aligned} \tag{28}
$$

Now we are ready to construct $X'$. Let

$$X'_{ik} := Y^\mu_{ik} - \Delta_{ik} + \frac{1}{n}\sum_{i'=1}^n \Delta_{i'k}. \tag{29}$$

In order for this to be a valid allocation, we need that $X'_{ik} \geq 0$, which is true because we showed above that $\Delta_{ik} \leq Y^\mu_{ik}$. We also need that $\sum_{i=1}^n X'_{ik} = 1$, which follows from

$$\sum_{i=1}^n X'_{ik} = \sum_{i=1}^n \left(Y^\mu_{ik} - \Delta_{ik} + \frac{1}{n}\sum_{i'=1}^n \Delta_{i'k}\right) = 1 - \sum_{i=1}^n \Delta_{ik} + \sum_{i'=1}^n \Delta_{i'k} = 1.$$

Next, we will show that Equation (23) is satisfied for all $i$ for fractional allocation $X'$. Starting with the left hand side of Equation (23), we have

$$X'_i \cdot \mu_i - \frac{1}{n}\|\mu_i\|_1$$

$$= Y_i^\mu \cdot \mu_i - \Delta_i \cdot \mu_i + \sum_{k=1}^{m} \frac{1}{n} \sum_{i'=1}^{n} \Delta_{i'k}\mu_{ik} - \frac{1}{n}\|\mu_i\|_1 \qquad \text{[Eq (29)]}$$

$$= S_i - \Delta_i \cdot \mu_i + \frac{1}{n} \sum_{i'=1}^{n} \Delta_{i'} \cdot \mu_i \qquad \text{[Eq (24)]}$$

$$\geq \frac{1}{n} \sum_{i'=1}^{n} \Delta_{i'} \cdot \mu_i \qquad \text{[Eq (28)]}$$

$$\geq \frac{a}{n} \sum_{i'=1}^{n} \sum_{k=1}^{m} \Delta_{i'k} \qquad [\mu_{ik} \geq a]$$

$$= \gamma. \qquad \text{[Eq (26)]}$$

Furthermore, we can bound the decrease in social welfare between fractional allocation $X'$ and $Y^\mu$ by

$$\langle Y^\mu, \mu \rangle_F - \langle X', \mu \rangle_F = \langle Y^\mu - X', \mu \rangle_F$$

$$\leq \sum_{i=1}^{n} \Delta_i \cdot \mu_i \qquad \text{[Equation (29)]}$$

$$\leq b \sum_{i=1}^{n} \sum_{k=1}^{m} \Delta_{ik} \qquad [\mu_{ik} \leq b]$$

$$\leq \frac{b}{a} n\gamma. \qquad \text{[Equation (26)]}$$

Therefore, we have shown that $X'$ has the desired properties, and thus the proportionality constraints satisfy Property 2. $\square$

## F  Proof of Lemma 1

For Section F only, we will assume w.l.o.g. that $a = 1$ and that $b \geq 1$. This is without loss of generality because envy-freeness in expectation constraints and social welfare are both scale invariant. Therefore, scaling every player's values (and mean values) by $1/a$ will give an equivalent problem where $a = 1$.

To prove Lemma 1, will show that we can transform $Y^\mu$ into a fractional allocation $X'$ which satisfies Property 2 through Algorithm 3. Algorithm 3 iterates over the following types of 'envy-with-slack' graphs, which track whether a player prefers their allocation by at least $\alpha$ over another player's allocation.

**Definition 6.** Let create-slack-graph$(\mu, X, \alpha)$ be the subroutine which returns the directed graph with vertices $N$ and edges generated as follows. Suppose $G = $ create-slack-graph$(\mu, X, \alpha)$. Then a directed edge from $i$ to $i'$ exists in $G$ if and only if

$$\langle X_i, \mu_i \rangle - \langle X_{i'}, \mu_i \rangle < \alpha.$$

At a high level, Algorithm 3 constructs 'envy-with-slack' graphs with progressively smaller $\alpha$, with $\alpha \geq \gamma$ for all iterations. The algorithm operates on sets of nodes called *equivalence classes*, where every pair of nodes in an equivalence class has the same allocation. We represent the set of equivalence classes in a fractional allocation $X$ as $\mathcal{S}(X)$.

**Definition 7.** Let $\mathcal{S}(X)$ be the set of equivalence classes of fractional allocation $X$, where two nodes $i, i'$ are part of the same equivalence class $S \in \mathcal{S}(X)$ if and only if $X_i = X_{i'}$.

Algorithm 3 generally makes progress by either 1) merging two equivalence classes, or 2) removing an edge from the graph. We formalize this model below.

Each iteration $r$ begins with some allocation $X^r$ and a slack value $\alpha^r$. Algorithm 3 then generates from these parameters a directed graph $G^r = \text{create-slack-graph}(\mu, X^r, \alpha^r)$, which is the 'envy-with-slack' graph for allocation $X^r$ given means $\mu$. As in standard graph notation, for a graph $G$ we define $V(G)$ as the vertices of $G$ and $E(G)$ as the edges of $G$. Each edge $e = (i, i') \in E(G^r)$ has a weight $w_e = \langle X_i, \mu_i \rangle - \langle X_{i'}, \mu_i \rangle$. For a set of vertices $S$, we use $\delta^+(S)$ to represent the edges with head in $S$ and tail in $N \backslash S$, and similarly, we use $\delta^-(S)$ to represent the edges with tail in $S$ and head in $N \backslash S$. For notational convenience, we let $\delta^-(i) = \delta^-(\{i\})$, and $\delta^+(i) = \delta^+(\{i\})$. Throughout this section, we will use the notation $\text{sw}(X, \mu) = \langle X, \mu \rangle_F$.

An overview of the algorithm is as follows. In each iteration, Algorithm 3 generally performs one of three operations and removes an edge by decreasing $\alpha$. First, if there exists an equivalence class $S$ with at least one incoming edge but no outgoing edges, then operation remove-incoming-edge transfers allocation probability from nodes in $S$ to all other nodes. If such an equivalence class does not exist, then Algorithm 3 finds a specific type of cycle in the 'envy-with-slack' graph. If there exists some node which has an edge to some but not all of the nodes in the cycle, then operation cycle-shift gives each node in the cycle half of its current allocation and half of the next node's allocation. If such a node does not exist and all edges in the graph have low enough weight, then operation average-clique instead creates a new equivalence class by merging all equivalence classes that the nodes in the cycle belong to. Such a merge may lead to envy, which is removed by a call to Algorithm 4. We define each of the three operations formally below, where each operation returns a new allocation $X'$.

**Definition 8.** Let $S \in \mathcal{S}(X)$. Define $X_{Sk} = X_{ik}$ for every $i \in S$. Let remove-incoming-edge$(S, \alpha, X)$ be the subroutine which returns $X'$, where

$$X'_{ik} = X_{ik} - \frac{(n - |S|)\alpha X_{Sk}}{2bn \sum_{k'} X_{Sk'}} \quad \forall i \in S$$

and

$$X'_{ik} = X_{ik} + \frac{|S|\alpha X_{Sk}}{2bn \sum_{k'} X_{Sk'}} \quad \forall i \in N \backslash S$$

**Definition 9.** Let $C$ be a cycle in a graph $G = \text{create-slack-graph}(\mu, X, \alpha)$ and $\text{next}(i)$ be the node which $i$ points to in $C$. Then the subroutine cycle-shift$(C, X)$ returns $X'$, where

$$X'_{ik} = \frac{X_{ik} + X_{\text{next}(i)k}}{2} \quad \forall i \in V(C)$$
$$X'_{ik} = X_{ik} \quad \forall i \in N \backslash V(C)$$

**Definition 10.** Let $Q$ be a clique in a graph $G = \text{create-slack-graph}(\mu, X, \alpha)$. Then the subroutine average-clique$(Q, X)$ returns $X'$, where

$$X'_{ik} = \frac{\sum_{i' \in Q} X_{i'k}}{|Q|} \quad \forall i \in Q$$
$$X'_{ik} = X_{ik} \quad \forall i \in N \backslash Q$$

We also define two intermediary operations. Note that the $\arg\min$ function may return the empty set, a singleton, or a set with multiple elements.

**Definition 11.** Let $G = \text{create-slack-graph}(\mu, X, \alpha)$ with edge set $E$. For each equivalence class $S \in \mathcal{S}(X)$, let $E_S = \arg\min_{e \in \delta^+(S)} w_e$, where the size of $E_S$ may be $0, 1$, or $> 1$. Let $E' = \sum_S E_S$, and let $G' = (N, E')$. Then the subroutine find-special-cycle$(G, \mathcal{S}(X))$ returns a cycle $C$ of $G'$ where $V(C)$ contains at most one member of each equivalence class $S \in \mathcal{S}(X)$ (and returns $\emptyset$ if no such cycle exists).

**Definition 12.** Let $G$ = create-slack-graph$(\mu, X, \alpha)$ and let $U \subset N$. Let distribute-equally$(U, \beta, X)$ be the subroutine which returns $X'$, where

$$X'_{ik} = (1 - |N \backslash U| \cdot \beta) \cdot X_{ik} \quad \forall\, i \in U$$

$$X'_{ik} = X_{ik} + \beta \cdot \sum_{i' \in U} X_{i'k} \quad \forall\, i \in N \backslash U$$

We are now ready to present Algorithm 3.

---

**Algorithm 3** Achieving Envy with Slack

---

Require $Y^\mu, \mu$
Let $\alpha^0 \leftarrow \gamma \cdot (e^{n^2 \log(4bn^4) + n^3 \log(2n)}, X^0 \leftarrow Y^\mu, G^0 \leftarrow$ create-slack-graph$(\mu, X^0, \alpha^0), r \leftarrow 0$.
**while** $\exists (u, v) \in E(G^r)$ s.t. $u, v$ are in different equivalence classes **do**
 **if** $\exists\, S \in \mathcal{S}(X^r)$ s.t. $\delta^-(S) \neq \emptyset$ and $\delta^+(S) = \emptyset$ **then**
  $\triangleright X^{r+1} =$ remove-incoming-edge$(S, \alpha^r, X^r)$
  $\triangleright \alpha^{r+1} = \frac{\alpha^r}{2b}$
 **else**
  $\triangleright C =$ find-special-cycle$(G^r, \mathcal{S}(X^r))$
  **if** $\exists\, u \in N$ and $\exists\, i, i' \in V(C)$ such that $(u, i) \in E$ and $(u, i') \notin E$ **then**
   $\triangleright X^{r+1} =$ cycle-shift$(C, X^r)$
   $\triangleright \alpha^{r+1} = \frac{\alpha^r}{2}$
  **else if** $\exists\, e \in E(G^r)$ s.t. $w_e \geq \frac{\alpha^r}{4bn^4}$ **then**
   $\triangleright X^{r+1} = X^r$
   $\triangleright \alpha^{r+1} = \frac{\alpha^r}{4bn^4}$
  **else**
   $\triangleright \mathcal{S}' = \{S \in \mathcal{S}(X^r) : S \cap V(C) \neq \emptyset\}$
   $\triangleright Q = \bigcup_{S \in \mathcal{S}'} S$
   $\triangleright X^{\mathsf{avg}} =$ average-clique$(Q, X^r)$
   $\triangleright \alpha^{\mathsf{avg}} = \frac{\alpha^r}{n}$
   $\triangleright G^{\mathsf{avg}} =$ create-slack-graph$(\mu, X^{\mathsf{avg}}, \alpha^{\mathsf{avg}})$ // $G^{\mathsf{avg}}$ *defined for analysis only.*
   $\triangleright X' = X^{\mathsf{avg}}$
   $\triangleright G' =$ create-slack-graph$(\mu, X', 0)$
   **while** $\exists e \in E(G')$ **do**
    $\triangleright X' =$ remove-envy$(\mu, X')$
    $\triangleright G' =$ create-slack-graph$(\mu, X', 0)$
   **end while**
   $\triangleright X^{r+1} = X'$
   $\triangleright \alpha^{r+1} = \frac{\alpha^{\mathsf{avg}}}{2}$
  **end if**
 **end if**
 $\triangleright G^{r+1} \leftarrow$ create-slack-graph$(\mu, X^{r+1}, \alpha^{r+1})$
 $\triangleright r = r + 1$
**end while**
**return** $X^r$

---

Algorithm 3 calls remove-envy, which is equivalent to calling Algorithm 4. Algorithm 4 will require the following definition.

**Definition 13.** Let create-non-negative-envy-graph$(\mu, X)$ be the subroutine which returns the directed graph with vertices $N$ and edges generated as follows. Suppose $G$ = create-non-negative-envy-graph$(\mu, X)$. Then a directed edge from $i$ to $i'$ exists in $G$ if and only if

$$\langle X_i, \mu_i \rangle - \langle X_{i'}, \mu_i \rangle \leq 0.$$

Note that this definition is exactly the same as that of create-slack-graph$(\mu, X, 0)$, except with a weak instead of a strict inequality. We now present Algorithm 4.

We begin by proving some helpful lemmas. It will be convenient to define the following term.

**Algorithm 4** Remove Envy (also referred to as subroutine remove-envy($\mu, X^0$))

---

Require $\mu, X^0$
Let $G^0 \leftarrow$ create-non-negative-envy-graph($\mu, X^0$), $r \leftarrow 0$.
**if** $\left|\{e \in E(G^0) : w_e < 0\}\right| = 0$ **then**
    **return** $X^0$
**end if**
$\triangleright$ Choose $(u, v) \in \{e \in E(G^0) : w_e < 0\}$
$\triangleright U \leftarrow \{v' \in E(G^0) : w_{(u,v')} < 0\}$
**while** $w_{(u,v)} < 0$ and there does not exist a cycle in $G^r$ containing $(u, v)$ **do**
    $\triangleright \forall\, i \in U, \beta_i = \min\limits_{i' \in N \backslash U, \beta \geq 0} \left(\beta \text{ s.t. distribute-equally}(U, \beta, X^r) \text{ returns } X' \text{ where } \langle X'_i, \mu_i \rangle = \langle X'_{i'}, \mu_i \rangle \right)$
    $\triangleright \beta_u = \beta$ s.t. distribute-equally($U, \beta, X^r$) returns $X'$ where $\langle X'_u, \mu_u \rangle = \langle X'_v, \mu_u \rangle$
    $\triangleright i^* = \arg\min_{i \in (\{u\} \cup U)} \beta_i$
    $\triangleright X^{r+1} = $ distribute-equally($U, \beta_{i^*}, X^r$)
    $\triangleright U = U \cup \{i^*\}$
    $\triangleright G^{r+1} \leftarrow$ create-non-negative-envy-graph($\mu, X^{r+1}$)
    $\triangleright r = r + 1$
**end while**
**if** $w_{(u,v)} < 0$ **then**
    $\triangleright C \leftarrow$ cycle in $G^r$ containing $(u, v)$
    // Let prev($i$) and next($i$) be the nodes before and after $i$ in $C$, respectively.
    $\triangleright \forall\, i \in V(C), X'_i = X^r_{\text{next}(i)}$
**else**
    $\triangleright X' = X^r$
**end if**
**return** $X'$

---

**Definition 14.** Let $X$ be an **envy-free** fractional allocation for $\mu$ if for all $i, i' \in N$,

$$\langle X_i, \mu_i \rangle - \langle X_{i'}, \mu_i \rangle \geq 0.$$

**Lemma 4.** *Let $X$ be an envy-free fractional allocation for $\mu$, and let $G = $ create-slack-graph($\mu, X, \alpha$) with edge set $E$. Suppose that there exists some equivalence class $S \in \mathcal{S}(X)$ such that $\delta^-(S) \neq \emptyset$ and $\delta^+(S) = \emptyset$ in $G$, and let $X' = $ remove-incoming-edge($S, \alpha, X$). Let $G' = $ create-slack-graph($\mu, X', \alpha/2b$) with edge set $E'$. Then $|E'| < |E|$.*

*proof.* We first show that if an edge $e \notin E$, then $e \notin E'$. Note that for any $i, i'$ such that $(i, i') \notin E$,

$$\langle X_i, \mu_i \rangle - \langle X_{i'}, \mu_i \rangle \geq \alpha > \frac{\alpha}{2b}.$$

Therefore, to show that an edge $(i, i')$ not in $E$ is also not in $E'$, it suffices to show that $\langle X'_i, \mu_i \rangle - \langle X'_{i'}, \mu_i \rangle \geq \langle X_i, \mu_i \rangle - \langle X_{i'}, \mu_i \rangle$.

The subroutine remove-incoming-edge($S, \alpha, X$) only transfers weight from $S$ to $N \backslash S$, which implies that no node in $N \backslash S$ will gain an edge to a node in $S$. Formally,

$$\langle X'_{i'}, \mu_{i'} \rangle - \langle X'_i, \mu_{i'} \rangle > \langle X_{i'}, \mu_{i'} \rangle - \langle X_i, \mu_{i'} \rangle \quad \forall\, i \in S, i' \in N \backslash S. \tag{30}$$

Every pair of nodes $i, i' \in S$ has their fractional allocation reduced by the same amount, so

$$\langle X'_i, \mu_i \rangle - \langle X'_{i'}, \mu_i \rangle = \langle X_i, \mu_i \rangle - \langle X_{i'}, \mu_i \rangle \quad \forall\, i, i' \in S. \tag{31}$$

Similarly, every pair of nodes $i, i' \in N \backslash S$ has their fractional allocation increased by the same amount, so

$$\langle X'_i, \mu_i \rangle - \langle X'_{i'}, \mu_i \rangle = \langle X_i, \mu_i \rangle - \langle X_{i'}, \mu_i \rangle \quad \forall\, i, i' \in N \backslash S. \tag{32}$$

Finally, we observe that for any $i \in S$ and $i' \in N \backslash S$,

$$\langle X'_i, \mu_i \rangle - \langle X'_{i'}, \mu_i \rangle = \left( \langle X_i, \mu_i \rangle - \sum_{k \in [m]} \frac{(n - |S|)\alpha X_{Sk}\mu_{ik}}{2bn \sum_{k'} X_{Sk'}} \right) - \left( \langle X_{i'}, \mu_i \rangle + \sum_{k \in [m]} \frac{|S|\alpha X_{Sk}\mu_{ik}}{2bn \sum_{k'} X_{Sk'}} \right)$$

$$= \langle X_i, \mu_i \rangle - \langle X_{i'}, \mu_i \rangle - \sum_{k \in [m]} \frac{n\alpha X_{Sk} \mu_{ik}}{2bn \sum_{k'} X_{Sk'}}$$

$$\geq \langle X_i, \mu_i \rangle - \langle X_{i'}, \mu_i \rangle - \sum_{k \in [m]} \frac{n\alpha X_{Sk}}{2n \sum_{k'} X_{Sk'}}$$

$$= \langle X_i, \mu_i \rangle - \langle X_{i'}, \mu_i \rangle - \frac{\alpha}{2}$$

$$\geq \alpha - \frac{\alpha}{2}$$

$$\geq \frac{\alpha}{2b}.$$

where the second inequality is because $(i, i') \notin E$. Therefore, by definition of $G'$, $(i, i') \notin E'$.

Recall that $\delta^-(S) \neq \emptyset$ in $G$. We will now show that $\delta^-(S) = \emptyset$ in $G'$. Observe that for $i \in S$ and $i' \in N \backslash S$,

$$\langle X'_{i'}, \mu_{i'} \rangle - \langle X'_i, \mu_{i'} \rangle = \left( \langle X_{i'}, \mu_{i'} \rangle + \sum_{k \in [m]} \frac{|S|\alpha X_{Sk} \mu_{i'k}}{2bn \sum_{k'} X_{Sk'}} \right) - \left( \langle X_i, \mu_{i'} \rangle - \sum_{k \in [m]} \frac{(n - |S|)\alpha X_{Sk} \mu_{i'k}}{2bn \sum_{k'} X_{Sk'}} \right)$$

$$= \langle X_{i'}, \mu_{i'} \rangle - \langle X_i, \mu_{i'} \rangle + \sum_{k \in [m]} \frac{n\alpha X_{Sk} \mu_{i'k}}{2bn \sum_{k'} X_{Sk'}}$$

$$\geq \langle X_{i'}, \mu_{i'} \rangle - \langle X_i, \mu_{i'} \rangle + \frac{\alpha}{2b}$$

$$\geq \frac{\alpha}{2b}.$$

Therefore, $(i', i) \notin E$, which implies $\delta^-(S) = \emptyset$ in $G'$. We conclude that all edges in $E'$ exist in $E$, and at least one edge in $E$ does not exist in $E'$, which implies that $|E'| < |E|$. $\qquad\square$

**Lemma 5.** *Let* $X$ *be an envy-free fractional allocation for* $\mu$, *and let* $G =$ create-slack-graph$(\mu, X, \alpha)$ *with edge set* $E$. *Suppose that there exists some equivalence class* $S \in \mathcal{S}(X)$ *such that* $\delta^-(S) \neq \emptyset$ *and* $\delta^+(S) = \emptyset$ *in* $G$, *and let* $X' =$ remove-incoming-edge$(S, \alpha, X)$. *Then* $X'$ *is an envy-free allocation for* $\mu$.

*proof.* Let $G' =$ create-slack-graph$(\mu, X', \alpha/2b)$ with edge set $E'$. It suffices to show that $w_e \geq 0 \ \forall e \in E'$. By Lemma 4, if an edge $(i, i') \notin E$, then

$$\langle X'_i, \mu_i \rangle - \langle X'_{i'}, \mu_i \rangle \geq \tfrac{\alpha}{2b} \geq 0.$$

Consider any $i \in S$ and $i' \in N \backslash S$. Then there must not exist an edge $(i, i') \in E$ by assumption. Also by assumption, for any $i, i'$ such that $(i, i') \in E$, we have that $\langle X_i, \mu_i \rangle - \langle X_{i'}, \mu_i \rangle \geq 0$. By a direct application of Equations (30), (31), and (32), we can conclude that $\langle X'_i, \mu_i \rangle - \langle X'_{i'}, \mu_i \rangle \geq 0$ as well. $\qquad\square$

**Lemma 6.** *Let* $G =$ create-slack-graph$(\mu, X, \alpha)$ *with edge set* $E$. *Suppose that there exists some equivalence class* $S \in \mathcal{S}(X)$ *such that* $\delta^-(S) \neq \emptyset$ *and* $\delta^+(S) = \emptyset$ *in* $G$, *and let* $X' =$ remove-incoming-edge$(S, \alpha, X)$. *Then*

$$\mathrm{sw}(X, \mu) - \mathrm{sw}(X', \mu) \leq \frac{\alpha n}{2}.$$

*proof.* Observe that

$$\mathrm{sw}(X', \mu) = \sum_{i \in S} \langle X'_i, \mu_i \rangle + \sum_{i \in N \backslash S} \langle X'_i, \mu_i \rangle$$

$$\geq \sum_{i \in S} \left( 1 - \frac{(n - |S|)\alpha}{2bn \sum_{k'} X_{Sk'}} \right) \cdot \langle X_i, \mu_i \rangle + \sum_{i \in N \backslash S} \langle X_i, \mu_i \rangle$$

$$\geq \sum_{i \in S} \left(1 - \frac{\alpha}{2b \sum_{k'} X_{Sk'}}\right) \cdot \langle X_i, \mu_i \rangle + \sum_{i \in N \setminus S} \langle X_i, \mu_i \rangle$$

$$= \sum_{i \in [N]} \langle X_i, \mu_i \rangle - \frac{\alpha}{2b \sum_{k'} X_{Sk'}} \sum_{i \in S} \langle X_i, \mu_i \rangle$$

$$= \sum_{i \in [N]} \langle X_i, \mu_i \rangle - \frac{\alpha}{2b \sum_{k'} X_{Sk'}} \sum_{i \in S} \sum_k X_{Sk} \mu_{ik}$$

$$\geq \sum_{i \in [N]} \langle X_i, \mu_i \rangle - \frac{\alpha}{2 \sum_{k'} X_{Sk'}} \sum_{i \in S} \sum_k X_{Sk}$$

$$= \mathrm{sw}(X, \mu) - \frac{\alpha |S|}{2}$$

$$\geq \mathrm{sw}(X, \mu) - \frac{\alpha n}{2}.$$

This implies that

$$\mathrm{sw}(X, \mu) - \mathrm{sw}(X', \mu) \leq \frac{\alpha n}{2}.$$

$\square$

**Lemma 7.** *Let $X$ be an envy-free fractional allocation for $\mu$ and let $G = \mathsf{create\text{-}slack\text{-}graph}(\mu, X, \alpha)$ with edge set $E$. Let $C = \mathsf{find\text{-}special\text{-}cycle}(G, \mathcal{S}(X))$ and suppose there exists a node $u$ such that for $i, i' \in V(C)$, $(u, i) \in E$ and $(u, i') \notin E$. Let $X' = \mathsf{cycle\text{-}shift}(C, X)$ and let $G' = \mathsf{create\text{-}slack\text{-}graph}(\mu, X', \alpha/2)$ with edge set $E'$. Then $|E'| < |E|$.*

*proof.* We first show that if an edge $e \notin E$, then $e \notin |E'|$. Suppose that $i \in V(C), i' \in N$, and edge $(i, i') \notin E$. Then

$$\langle X'_i, \mu_i \rangle - \langle X'_{i'}, \mu_i \rangle = \frac{1}{2} \langle X_i, \mu_i \rangle + \frac{1}{2} \langle X_{\mathsf{next}(i)}, \mu_i \rangle - \langle X_{i'}, \mu_i \rangle$$

$$\geq \frac{1}{2} \langle X_i, \mu_i \rangle + \frac{1}{2} \langle X_{i'}, \mu_i \rangle - \langle X_{i'}, \mu_i \rangle$$

$$= \frac{1}{2} (\langle X_i, \mu_i \rangle - \langle X_{i'}, \mu_i \rangle) \tag{33}$$

$$\geq \frac{\alpha}{2}.$$

Now, suppose that $i, i' \notin V(C)$ and edge $(i, i') \notin E$. Then

$$\langle X'_i, \mu_i \rangle - \langle X'_{i'}, \mu_i \rangle = \langle X_i, \mu_i \rangle - \langle X_{i'}, \mu_i \rangle \geq \alpha. \tag{34}$$

Finally, suppose that $i \in V(C), i' \in N \setminus V(C)$, and edge $(i', i) \notin E$. Then

$$\langle X'_{i'}, \mu_{i'} \rangle - \langle X'_i, \mu_{i'} \rangle = \langle X_{i'}, \mu_{i'} \rangle - \frac{1}{2} \langle X_i, \mu_{i'} \rangle - \frac{1}{2} \langle X_{\mathsf{next}(i)}, \mu_{i'} \rangle$$

$$= \frac{1}{2} (\langle X_{i'}, \mu_{i'} \rangle - \langle X_i, \mu_{i'} \rangle) + \frac{1}{2} (\langle X_{i'}, \mu_{i'} \rangle - \langle X_{\mathsf{next}(i)}, \mu_{i'} \rangle) \tag{35}$$

$$\geq \frac{1}{2} (\alpha) + \frac{1}{2} (0)$$

$$= \frac{\alpha}{2}.$$

We have shown that if $e \in E'$, then $e \in E$. Now we will show that there exists at least one edge $e$ such that $e \in E$, but $e \notin E'$. Consider the node $u$ described in the lemma statement, and let $i \in V(C)$ be some node such that $(u, i) \in E$ but $(u, \mathsf{next}(i)) \notin E$. Suppose that $u \notin V(C)$. Then

$$\langle X'_u, \mu_u \rangle - \langle X'_i, \mu_u \rangle = \langle X_u, \mu_u \rangle - \frac{1}{2} \langle X_i, \mu_u \rangle - \frac{1}{2} \langle X_{\mathsf{next}(i)}, \mu_u \rangle$$

$$= \frac{1}{2} (\langle X_u, \mu_u \rangle - \langle X_i, \mu_u \rangle) + \frac{1}{2} (\langle X_u, \mu_u \rangle - \langle X_{\mathsf{next}(i)}, \mu_u \rangle)$$

$$\geq \frac{1}{2}(\alpha) + \frac{1}{2}(0)$$
$$= \frac{\alpha}{2}.$$

Suppose that $u \in V(C)$. Then

$$\langle X'_u, \mu_u \rangle - \langle X'_i, \mu_u \rangle = \frac{1}{2}\langle X_u, \mu_u \rangle + \frac{1}{2}\langle X_{\mathsf{next}(u)}, \mu_u \rangle - \frac{1}{2}\langle X_i, \mu_u \rangle - \frac{1}{2}\langle X_{\mathsf{next}(i)}, \mu_u \rangle$$
$$= \frac{1}{2}(\langle X_u, \mu_u \rangle - \langle X_i, \mu_u \rangle) + \frac{1}{2}(\langle X_{\mathsf{next}(u)}, \mu_u \rangle - \langle X_{\mathsf{next}(i)}, \mu_u \rangle)$$
$$\geq \frac{1}{2}(\alpha) + \frac{1}{2}(0)$$
$$= \frac{\alpha}{2}.$$

This implies that $(u, i) \notin E'$, as desired. $\qquad\square$

**Lemma 8.** *Let $X$ be an envy-free fractional allocation for $\mu$ and let $G = \mathsf{create\text{-}slack\text{-}graph}(\mu, X, \alpha)$ with edge set $E$. Let $C = \mathsf{find\text{-}special\text{-}cycle}(G, \mathcal{S}(X))$ and let $X' = \mathsf{cycle\text{-}shift}(C, X)$. Then $X'$ is an envy-free allocation for $\mu$.*

*proof.* Let $G' = \mathsf{create\text{-}slack\text{-}graph}(\mu, X', \alpha/2)$ with edge set $E'$. It suffices to show that $w_e \geq 0$ for all $e \in E'$. By Lemma 7, if an edge $(i, i') \notin E$, then

$$\langle X'_i, \mu_i \rangle - \langle X'_{i'}, \mu_i \rangle \geq \tfrac{\alpha}{2} \geq 0.$$

Recall that by assumption, for any $i, i'$ such that $(i, i') \in E$, $\langle X_i, \mu_i \rangle - \langle X_{i'}, \mu_i \rangle \geq 0$. Suppose that $i \in V(C), i' \in N$, and $(i, i') \in E$. Then by Equation (33),

$$\langle X'_i, \mu_i \rangle - \langle X'_{i'}, \mu_i \rangle \geq \frac{1}{2}(\langle X_i, \mu_i \rangle - \langle X_{i'}, \mu_i \rangle) \geq 0.$$

Suppose instead that $i, i' \notin V(C)$ and edge $(i, i') \in E$. Then by Equation (34),

$$\langle X'_i, \mu_i \rangle - \langle X'_{i'}, \mu_i \rangle = \langle X_i, \mu_i \rangle - \langle X_{i'}, \mu_i \rangle \geq 0.$$

Finally, suppose that $i \in V(C), i' \in N \backslash V(C)$, and edge $(i', i) \in E$. Then by Equation (35),

$$\langle X'_{i'}, \mu_{i'} \rangle - \langle X'_i, \mu_{i'} \rangle = \frac{1}{2}(\langle X_{i'}, \mu_{i'} \rangle - \langle X_i, \mu_{i'} \rangle) + \frac{1}{2}(\langle X_{i'}, \mu_{i'} \rangle - \langle X_{\mathsf{next}(i)}, \mu_{i'} \rangle)$$
$$\geq \frac{1}{2}(0) + \frac{1}{2}(0)$$
$$= 0.$$

$\qquad\square$

**Lemma 9.** *Let $X$ be an envy-free fractional allocation for $\mu$ and let $G = \mathsf{create\text{-}slack\text{-}graph}(\mu, X, \alpha)$ with edge set $E$. Let $C = \mathsf{find\text{-}special\text{-}cycle}(G, \mathcal{S}(X))$ and let $X' = \mathsf{cycle\text{-}shift}(C, X)$. Then*

$$\mathsf{sw}(X, \mu) - \mathsf{sw}(X', \mu) \leq \frac{n\alpha}{2}.$$

*proof.* Observe that

$$\mathsf{sw}(X', \mu) = \sum_{i \in V(C)} \langle X'_i, \mu_i \rangle + \sum_{i \in N \backslash V(C)} \langle X'_i, \mu_i \rangle$$
$$= \frac{1}{2} \cdot \sum_{i \in V(C)} (\langle X_i, \mu_i \rangle + \langle X_{\mathsf{next}(i)}, \mu_i \rangle) + \sum_{i \in N \backslash V(C)} \langle X_i, \mu_i \rangle$$
$$\geq \frac{1}{2} \cdot \sum_{i \in V(C)} (\langle X_i, \mu_i \rangle + \langle X_i, \mu_i \rangle - \alpha) + \sum_{i \in N \backslash V(C)} \langle X_i, \mu_i \rangle$$

$$= \sum_{i \in V(C)} \left( \langle X_i, \mu_i \rangle - \frac{\alpha}{2} \right) + \sum_{i \in N \setminus V(C)} \langle X_i, \mu_i \rangle$$

$$\geq \sum_{i \in V(C)} \left( \langle X_i, \mu_i \rangle - \frac{\alpha}{2} \right) + \sum_{i \in N \setminus V(C)} \left( \langle X_i, \mu_i \rangle - \frac{\alpha}{2} \right)$$

$$= \sum_{i \in N} \langle X_i, \mu_i \rangle - \frac{n\alpha}{2}$$

This implies that

$$\mathrm{sw}(X, \mu) - \mathrm{sw}(X', \mu) \leq \frac{n\alpha}{2}.$$

$\square$

**Lemma 10.** *Let $X$ be an envy-free fractional allocation for $\mu$. Let $G = \mathsf{create\text{-}slack\text{-}graph}(\mu, X, \alpha)$ with edge set $E$, and let $Q$ be a clique in $G$. Let $X' = \mathsf{average\text{-}clique}(Q, X)$ and $G' = \mathsf{create\text{-}slack\text{-}graph}(\mu, X', \alpha/n)$, with $E'$ the edge set of $G'$. Then $|E'| \leq |E|$.*

*proof.* It suffices to show that if an edge $e \notin E$, then $e \notin |E'|$. If $i, i' \in Q$, then edge $(i, i')$ must be in $E$, as $Q$ is a clique. Suppose that $i, i' \in N \setminus Q$ and $(i, i') \notin E$. Then

$$\langle X'_i, \mu_i \rangle - \langle X'_{i'}, \mu_i \rangle = \langle X_i, \mu_i \rangle - \langle X_{i'}, \mu_i \rangle \geq \alpha.$$

Now, suppose that $i \in Q, i' \in N \setminus Q$, and $(i, i') \notin E$. Then

$$\langle X'_i, \mu_i \rangle - \langle X'_{i'}, \mu_i \rangle = \frac{1}{|Q|} \left( \sum_{i'' \in Q} \langle X_{i''}, \mu_i \rangle \right) - \langle X_{i'}, \mu_i \rangle$$

$$\geq \langle X_i, \mu_i \rangle - \alpha \left( \frac{|Q| - 1}{|Q|} \right) - \langle X_{i'}, \mu_i \rangle$$

$$\geq \alpha - \alpha \left( \frac{|Q| - 1}{|Q|} \right)$$

$$\geq \frac{\alpha}{n}.$$

Finally, suppose that $i \in Q, i' \in N \setminus Q$, and $(i', i) \notin E$. Then

$$\langle X'_{i'}, \mu_{i'} \rangle - \langle X'_i, \mu_{i'} \rangle = \langle X_{i'}, \mu_{i'} \rangle - \frac{1}{|Q|} \left( \sum_{i'' \in Q} \langle X_{i''}, \mu_{i'} \rangle \right)$$

$$= \frac{1}{|Q|} \left( \sum_{i'' \in Q} \langle X_{i'}, \mu_{i'} \rangle - \langle X_{i''}, \mu_{i'} \rangle \right)$$

$$= \frac{1}{|Q|} \left( \langle X_{i'}, \mu_{i'} \rangle - \langle X_i, \mu_{i'} \rangle + \sum_{\{i'' \in Q : i'' \neq i\}} \langle X_{i'}, \mu_{i'} \rangle - \langle X_{i''}, \mu_{i'} \rangle \right)$$

$$\geq \frac{1}{n}(\alpha + 0)$$

$$\geq \frac{\alpha}{n}.$$

$\square$

**Lemma 11.** *Let $X$ be an envy-free fractional allocation for $\mu$. Let $G = \mathsf{create\text{-}slack\text{-}graph}(\mu, X, \alpha)$ with edge set $E$, and let $Q$ be a clique in $G$. Let $X' = \mathsf{average\text{-}clique}(Q, X)$. Then*

$$\langle X'_i, \mu_i \rangle - \langle X'_{i'}, \mu_i \rangle \geq -\alpha \quad \forall i, i' \in N.$$

*proof.* First, suppose $i' \in N \backslash Q$ and $i \in N$. Then

$$\langle X'_{i'}, \mu_{i'} \rangle - \langle X'_i, \mu_{i'} \rangle \geq \min_{i'' \in N} \langle X_{i'}, \mu_{i'} \rangle - \langle X_{i''}, \mu_{i'} \rangle \geq 0.$$

Suppose instead that $i, i' \in Q$. Then

$$\langle X'_{i'}, \mu_{i'} \rangle - \langle X'_i, \mu_{i'} \rangle = 0.$$

Finally, suppose that $i \in Q, i' \in N \backslash Q$. Then

$$\begin{aligned}
\langle X'_i, \mu_i \rangle - \langle X'_{i'}, \mu_i \rangle &= \frac{1}{|Q|} \left( \sum_{i'' \in Q} \langle X_{i''}, \mu_i \rangle \right) - \langle X_{i'}, \mu_i \rangle \\
&\geq \langle X_i, \mu_i \rangle - \alpha \left( \frac{|Q| - 1}{|Q|} \right) - \langle X_{i'}, \mu_i \rangle \\
&\geq 0 - \alpha \left( \frac{|Q| - 1}{|Q|} \right) \\
&\geq -\alpha.
\end{aligned}$$

$\square$

**Lemma 12.** *Let $G = \mathsf{create\text{-}slack\text{-}graph}(\mu, X, \alpha)$ with edge set $E$, and let $Q$ be a clique in $G$. Let $X' = \mathsf{average\text{-}clique}(Q, X)$. Then*

$$\mathrm{sw}(X, \mu) - \mathrm{sw}(X', \mu) \leq n \cdot \alpha.$$

*proof.* Observe that

$$\begin{aligned}
\mathrm{sw}(X', \mu) &= \sum_{i \in Q} \langle X'_i, \mu_i \rangle + \sum_{i \in N \backslash Q} \langle X'_i, \mu_i \rangle \\
&\geq \sum_{i \in Q} (\langle X_i, \mu_i \rangle - \alpha) + \sum_{i \in N \backslash Q} \langle X_i, \mu_i \rangle \\
&\geq \sum_{i \in N} (\langle X_i, \mu_i \rangle - \alpha) \\
&\geq \mathrm{sw}(X, \mu) - n \cdot \alpha.
\end{aligned}$$

Rearranging, this implies that

$$\mathrm{sw}(X, \mu) - \mathrm{sw}(X', \mu) \leq n \cdot \alpha.$$

$\square$

**Lemma 13.** *Let $G = \mathsf{create\text{-}slack\text{-}graph}(\mu, X, 0)$ with edge set $E$, and let $X' = \mathsf{remove\text{-}envy}(\mu, X)$. Further let $G' = \mathsf{create\text{-}slack\text{-}graph}(\mu, X', 0)$ with edge set $E'$. Then $|E'| < |E|$.*

*proof.* First, we show that if an edge $e \notin E$, then $e \notin E'$. In other words, we will show that no new edges with positive envy (negative weight) are added. Observe that within the while loop, remove-envy distributes $\beta_{i*}$ from a set $U$ to $N \backslash U$. If at the beginning of the while loop $\exists i \in U$ such that $w_{i,i'} < 0$ for some $i' \in N \backslash U$, then $\beta_{i*} = 0$ and $i'$ is added to $U$. Otherwise, by definition $\beta_{i*}$ is at most the minimum fraction that the set $U$ needs to give away in order to create a new 0 envy edge between any node in $U$ and a node $i' \in N \backslash U$. Therefore, distribute-equally cannot create a new edge with positive envy by our choice of $\beta_{i*}$, which implies that no new edge with positive envy could have been created by the end of the while loop. Now, suppose that $w_{(u,v)} < 0$ at the end of the while loop. Then there is a cycle $C$ in $G^\tau$ containing $(u, v)$. Then for every $i$, $\langle X'_i, \mu_i \rangle \geq \langle X^\tau_i, \mu_i \rangle$. The total set of allocations has not changed, so no positive envy is introduced.

Now, we show that there exists an edge $e \in E$ such that $e \notin E'$. That is, we show that we have removed some edge with positive envy. If $w_{(u,v)} = 0$ at the end of the while loop, then $(u, v) \in E$

and $(u, v) \notin E'$ so we are done. Otherwise, there is a cycle in $G^r$ containing $(u, v)$. As the overall set of allocations has not changed and $\langle X_v, \mu_u \rangle - \langle X_u, \mu_u \rangle > 0$, we must have

$$\left| i \in V(C) \text{ s.t. } \langle X_i', \mu_u \rangle - \langle X_u', \mu_u \rangle < 0 \right| < \left| i \in V(C) \text{ s.t. } \langle X_i, \mu_u \rangle - \langle X_u, \mu_u \rangle < 0 \right|.$$

Therefore, there exists some edge $e = (u, i)$ for $i \in V(C)$ such that $e \in E$ but $e \notin E'$. $\qquad \square$

**Lemma 14.** *Let* $G = \mathsf{create\text{-}slack\text{-}graph}(\mu, X, 0)$ *with edge set* $E$. *Suppose that*

$$-\tfrac{\alpha}{4bn^3} \leq \langle X_i, \mu_i \rangle - \langle X_{i'}, \mu_i \rangle \quad \forall i, i' \in N.$$

*Let* $X^0 = X$ *and define* $X^\ell = \mathsf{remove\text{-}envy}(\mu, X^{\ell-1})$. *Then for all* $i, i', \ell,$

$$-\tfrac{\alpha}{4n^2} \leq \langle X_i^\ell, \mu_{i'} \rangle - \langle X_i^{\ell-1}, \mu_{i'} \rangle \leq \tfrac{\alpha}{4n^3}.$$

*proof.* We first prove by induction on $\ell$ that for all $\ell$,

$$-\tfrac{\alpha}{4bn^3} \leq \langle X_i^\ell, \mu_i \rangle - \langle X_{i'}^\ell, \mu_i \rangle \quad \forall i, i' \in N.$$

The base case is true by assumption. Now consider any $\ell$. By the inductive hypothesis, we have that for all $i, i' \in N$, $-\tfrac{\alpha}{4bn^3} \leq \langle X_i^{\ell-1}, \mu_i \rangle - \langle X_{i'}^{\ell-1}, \mu_i \rangle$. Suppose for contradiction that there exists some $i, i' \in N$ such that $-\tfrac{\alpha}{4bn^3} > \langle X_i^\ell, \mu_i \rangle - \langle X_{i'}^\ell, \mu_i \rangle$. This means that the envy of $i$ for $i'$ must have increased to more than during $\tfrac{\alpha}{4bn^3}$ in the $\ell^{th}$ call to remove-envy. The only way for the envy of $i$ for $i'$ to increase in remove-envy is if $i \in U$, $i' \in N \backslash U$, and $\beta_{i*} > 0$. If $\langle X_i^{\ell-1}, \mu_i \rangle - \langle X_{i'}^{\ell-1}, \mu_i \rangle \leq 0$, then $\beta_{i*} = 0$. Therefore, we must have $\langle X_i^{\ell-1}, \mu_i \rangle - \langle X_{i'}^{\ell-1}, \mu_i \rangle > 0$. However, by our choice of $\beta_{i*}$, $i'$ must then be added to $U$ before $i$ becomes envious of $i'$. Therefore, it is not possible for the envy of $i$ for $i'$ to have increased to more than $\tfrac{\alpha}{4bn^3}$ in the $\ell^{th}$ call to remove-envy.

We now prove the main lemma. Consider any $\ell$. One stopping condition of the while loop in remove-envy is when $w_{(u,v)} = \langle X_u^r, \mu_u \rangle - \langle X_v^r, \mu_u \rangle = 0$. Observe that $v \in U$ and $u \in N \backslash U$ for all iterations $r$ in the while loop. This implies that $u$'s allocation is always increasing, while $v$'s allocation is always decreasing. The increase in $u$'s utility is therefore at most $u$'s current envy towards $v$'s allocation in $X_i^{\ell-1}$, which is upper bounded by $\tfrac{\alpha}{4bn^3}$ by the induction proof above. Formally,

$$\langle X_u^\ell, \mu_u \rangle - \langle X_u^{\ell-1}, \mu_u \rangle \leq \langle X_v^{\ell-1}, \mu_u \rangle - \langle X_u^{\ell-1}, \mu_u \rangle \leq \frac{\alpha}{4bn^3}. \tag{36}$$

Furthermore, over the course of remove-envy, the allocation of node $u$ is increased at least as much as that of any other node $i$, i.e.

$$(X_{uk}^\ell - X_{uk}^{\ell-1}) \geq (X_{ik}^\ell - X_{ik}^{\ell-1}) \quad \forall i \in N, k \in [m].$$

This is because $u$ is always a member of $N \backslash U$. Therefore, for any $i, i' \in N$,

$$\langle X_i^\ell, \mu_{i'} \rangle - \langle X_i^{\ell-1}, \mu_{i'} \rangle \leq b \cdot \frac{\alpha}{4bn^3} = \frac{\alpha}{4n^3},$$

as $b$ is the largest possible utility ratio between two nodes.

Again by applying Equation (36) and because $b$ is the largest possible utility ratio between two nodes, for any $i, i' \in [N]$, the utility of $i'$ for the allocation transferred from $i$ to $u$ is at most $b \cdot \tfrac{\alpha}{4bn^3} = \tfrac{\alpha}{4n^3}$. Node $i$ could have transferred to at most $n$ nodes during remove-envy, which implies that node $i'$ has utility of at most $n \cdot \tfrac{\alpha}{4n^3} = \tfrac{\alpha}{4n^2}$ for all of the allocation transferred away from node $i$ during remove-envy. Therefore,

$$\langle X_i^\ell, \mu_{i'} \rangle - \langle X_i^{\ell-1}, \mu_{i'} \rangle \geq -\frac{\alpha}{4n^2} \quad \forall\, i \in N.$$

$\qquad \square$

**Lemma 15.** *Let* $G = \mathsf{create\text{-}slack\text{-}graph}(\mu, X, 0)$ *with edge set* $E$. *Suppose that*

$$-\tfrac{\alpha}{4bn^3} \leq \langle X_i, \mu_i \rangle - \langle X_{i'}, \mu_i \rangle \quad \forall i, i' \in N.$$

*Let* $X^0 = X$ *and define* $X^\ell = \mathsf{remove\text{-}envy}(\mu, X^{\ell-1})$. *Then for all* $i, i'$ *and for all* $\ell \leq n^2,$

$$\mathsf{sw}(X^0, \mu) - \mathsf{sw}(X^\ell, \mu) \leq \tfrac{n\alpha}{4}.$$

*proof.* Consider some $\ell$. By Lemma 14 we know that

$$\langle X_i^\ell, \mu_i \rangle - \langle X_i^{\ell-1}, \mu_i \rangle \geq -\frac{\alpha}{4n^2} \quad \forall i \in N.$$

Applying the above once for each node, we obtain

$$\begin{aligned}
\mathrm{sw}(X^\ell, \mu) = \sum_{i \in N} \langle X_i^\ell, \mu_i \rangle \\
\geq \sum_{i \in N} \left( \langle X_i^\ell, \mu_i \rangle - \frac{\alpha}{4n^2} \right) \\
\geq \mathrm{sw}(X^{\ell-1}, \mu) - n \cdot \frac{\alpha}{4n^2}.
\end{aligned}$$

Rearranging, this implies that

$$\mathrm{sw}(X^{\ell-1}, \mu) - \mathrm{sw}(X^\ell, \mu) \leq \frac{\alpha}{4n}.$$

Because $\ell \leq n^2$, we therefore must have

$$\mathrm{sw}(X^0, \mu) - \mathrm{sw}(X^\ell, \mu) \leq \frac{n\alpha}{4}.$$

$\square$

**Lemma 16.** *Let $G = \mathsf{create\text{-}slack\text{-}graph}(\mu, X, \alpha)$ with edge set $E$. Suppose that for $S \in \mathcal{S}(X)$, if $\delta^-(S) \notin \emptyset$, then $\delta^+(S) \notin \emptyset$. Further suppose that*

$$-\tfrac{\alpha}{4bn^3} \leq \langle X_i, \mu_i \rangle - \langle X_{i'}, \mu_i \rangle \quad \forall i, i' \in N$$

*Let $X^0 = X$ and define $X^\ell = \mathsf{remove\text{-}envy}(\mu, X^{\ell-1})$. Finally, for $\ell \leq n^2$ let $G' = \mathsf{create\text{-}slack\text{-}graph}(\mu, X^\ell, \frac{\alpha}{2})$ with edge set $E'$. Then $|E'| \leq |E|$.*

*proof.* It suffices to show that if an edge $(i, i') \notin E$, then $(i, i') \notin E'$. By Lemma 14, in each call to remove-envy, the utility of $i$ for the allocation of $i$ decreases by at most $\frac{\alpha}{4n^2}$. Therefore,

$$\langle X_i^\ell, \mu_i \rangle - \langle X_i^0, \mu_i \rangle \geq \ell \cdot -\frac{\alpha}{4n^2} \geq -\frac{\alpha}{4}.$$

Again by Lemma 14, in each call to remove-envy, the utility of $i$ for the allocation of $i'$ increases by at most $\frac{\alpha}{4n^3}$. Therefore,

$$\langle X_{i'}^\ell, \mu_i \rangle - \langle X_{i'}^0, \mu_i \rangle \leq \ell \cdot \frac{\alpha}{4n^3} \leq \frac{\alpha}{4n}.$$

Putting both equations together, we have

$$\begin{aligned}
\langle X_i^\ell, \mu_i \rangle - \langle X_{i'}^\ell, \mu_i \rangle &\geq \langle X_i^0, \mu_i \rangle - \frac{\alpha}{4} - \left( \langle X_{i'}^0, \mu_i \rangle + \frac{\alpha}{4n} \right) \\
&= \langle X_i^0, \mu_i \rangle - \langle X_{i'}^0, \mu_i \rangle - \frac{\alpha}{4} - \frac{\alpha}{4n} \\
&\geq \alpha - \frac{\alpha}{2} \\
&= \frac{\alpha}{2}
\end{aligned}$$

as desired. $\square$

**Lemma 17.** *Let $X' = \mathsf{remove\text{-}envy}(\mu, X)$. Then $|\mathcal{S}(X')| \leq |\mathcal{S}(X)|$.*

*proof.* It suffices to show that no equivalence class $S \in S(X)$ becomes smaller (i.e. strictly loses members) during remove-envy. First, suppose for contradiction that $S$ first becomes smaller during the while loop during iteration $r$. Then in iteration $r$, it must be the case that $S \cap U \neq \emptyset$ and $S \cap (N \backslash U) \neq \emptyset$, otherwise the allocation of each member of $S$ would have changed by the same amount. Furthermore, it must be the case that in iteration $r$, $\beta_{i^*} > 0$, otherwise no allocation would have been transferred. In iteration $r$, consider some $i \in (S \cap U)$ and $i' \in (S \cap (N \backslash U))$. Then $\beta_i = 0$,

as $i$ begins iteration $r$ with zero envy towards $i'$. Because $\beta_{i*} \leq \min_i \beta_i$, this means $\beta_{i*} \leq 0$, which is a contradiction . Therefore, no equivalence class $S$ becomes smaller during the while loop. To conclude the proof, we observe that in the cycle elimination step after the while loop, the total set of allocations remains the same, which implies that the set of equivalence classes remains the same as well. $\qquad\square$

We are finally ready to prove Lemma 1.

*Proof of Lemma 1.* We first prove by induction that every iteration $r$ starts with an envy-free allocation $X^r$. The base case is satisfied because $X^0 = Y^\mu$, and $Y^\mu$ is envy-free by definition. Now, suppose that the inductive hypothesis holds for all iterations up to and including $r$. We will show that iteration $r+1$ starts with an envy-free allocation. If $X^{r+1} = X^r$, then we can directly invoke the inductive hypothesis for iteration $r$. Otherwise, suppose that remove-incoming-edge was called in iteration $r$ of Algorithm 3. Then by Lemma 5, iteration $r+1$ starts with an envy-free allocation. Suppose instead that cycle-shift was called in iteration $r$. Then by Lemma 8, iteration $r+1$ starts with an envy-free allocation. Finally, suppose that average-clique was called in iteration $r$. Then the remove-envy is called repeatedly as long as there exists an edge with negative weight in $E(G')$. By Lemma 13, each call to remove-envy removes an edge with negative weight from $E(G')$, and adds no new edges with negative weight. There are a finite number of edges in $E(G')$, so this loop terminates. Therefore, iteration $r+1$ starts with an envy-free allocation.

Next, we prove that in every iteration, either an edge is removed from the envy-with-slack graph, or the number of equivalence classes decreases. Formally, for two iterations $r$ and $r+1$, we prove that either

1. $|E(G^r)| > |E(G^{r+1})|$ or
2. $|E(G^r)| \geq |E(G^{r+1})|$ and $|\mathcal{S}(X^r)| > |\mathcal{S}(X^{r+1})|$.

If remove-incoming-edge is called in iteration $r$ of Algorithm 3, then by Lemma 4 we have $|E(G^r)| > |E(G^{r+1})|$. If cycle-shift is called in iteration $r$, then by Lemma 7 we have $|E(G^r)| > |E(G^{r+1})|$. If $\exists\, e \in E(G^r)$ s.t. $w_e \geq \frac{\alpha}{4bn^4}$, then $e \in E(G^r)$ but $e \notin E(G^{r+1})$. Furthermore, if $e' \notin E(G^r)$, then $e' \notin E(G^{r+1})$ as $X^r = X^{r+1}$. Therefore, we have $|E(G^r)| > |E(G^{r+1})|$.

Finally, suppose average-clique is called in iteration $r$ on clique $Q$. Recall that $G^{\mathsf{avg}} =$ create-slack-graph$(\mu, \text{average-clique}(Q, X^r), \alpha^{\mathsf{avg}})$. By Lemma 10, we know that $|E(G^r)| \geq |E(G^{\mathsf{avg}})|$. The number of edges in $E(G^{\mathsf{avg}})$ is at most $n^2$, so remove-envy will be called at most $n^2$ times by Lemma 13. Because average-clique was called, we know that $w_e < \frac{\alpha^r}{4bn^4}$ $\forall e \in E(G^r)$. By Lemma 11 with $\alpha = \frac{\alpha^r}{4bn^4}$,

$$\langle X_i^{\mathsf{avg}}, \mu_i \rangle - \langle X_{i'}^{\mathsf{avg}}, \mu_i \rangle \geq -\frac{\alpha^r}{4bn^4} = -\frac{\alpha^{\mathsf{avg}}}{4bn^3} \quad \forall i, i' \in N.$$

Therefore, we can apply Lemma 16 with $\alpha = \alpha^{\mathsf{avg}}$ to conclude that $|E(G^{r+1})| \leq |E(G^{\mathsf{avg}})| \leq |E(G^r)|$. Finally, observe that because $C$ included members of at least two equivalence classes, the operation average-clique strictly decreased the number of equivalence classes. By Lemma 17, operation remove-envy does not increase the number of equivalence classes. Therefore, $|\mathcal{S}(X^r)| > |\mathcal{S}(X^{r+1})|$.

We now prove that the algorithm terminates with an $X^r$ which satisfies Proposition 2. Each iteration either removes an edge or merges two equivalence classes. Because the maximum number of edges is $n^2$ and the number of equivalence classes is $n$, Algorithm 3 terminates in at most $n^3$ iterations. We need to show that Algorithm 3 terminates with $\alpha^r \geq \gamma$. If an iteration $r$ does not call remove-envy, then an edge is removed and $\alpha^{r+1} \geq \frac{\alpha^r}{4bn^4}$. There can be at most $n^2$ such iterations. If an iteration $r$ does call remove-envy, then $\alpha^{r+1} = \frac{\alpha^r}{2n}$. There can be at most $n$ such iterations which call remove-envy between every iteration which does not call remove-envy, for a total of at most $n^3$ iterations. Therefore,

$$\alpha^r \geq \frac{\alpha_0}{(4bn^4)^{n^2} \cdot (2n)^{n^3}} = \frac{\alpha_0}{e^{n^2 \log(4bn^4) + n^3 \log(2n)}}.$$

Choosing $\alpha_0 = \gamma \cdot \left(e^{n^2 \log(4bn^4) + n^3 \log(2n)}\right)$ thus implies that $\alpha_r \geq \gamma$ for every iteration $r$ if $\gamma \cdot \left(e^{n^2 \log(4bn^4) + n^3 \log(2n)}\right) \leq 1$. Therefore, we set $\gamma_0 = e^{-n^2 \log(4bn^4) - n^3 \log(2n)}$.

Finally, we need to show that Algorithm 3 does not significantly decrease the social welfare. By Lemmas 6, 9 and 12, we know that each of the operations remove-incoming-edge, cycle-shift, and average-clique change the social welfare by at most $O(\gamma)$. Each of these operations is called at most $n^3$ times. Each of remove-incoming-edge and cycle-shift are called at most once for each edge, or at most $n^2$ times. Operation average-clique is called at most $n$ times for each edge, or at most $n^3$ times. Finally, Lemma 15 bounds the total social welfare loss from all calls to remove-envy between any two calls to average-clique by $O(\gamma)$. Therefore, $\mathrm{sw}(Y^\mu, \mu) - \mathrm{sw}(X^r, \mu) = O(\gamma)$, as desired. □

# G Proof of Theorem 2

*proof.* Let $a = 1$, $b = 3$, $n = 2$, and $m = 2$, and assume that the distributions of values are all normal distributions with variance 1. For $n = 2$, envy-freeness and proportionality are equivalent, and therefore we will focus on the former. Define $\epsilon = T^{-1/3}$. We will prove the desired result by contradiction. Assume there exists an algorithm ALG such that for any $\mu^* \in [a, b]^{2 \times 2}$, the probability that the algorithm satisfies the envy-freeness constraints and has regret of less than $\frac{T^{2/3}}{\log(T)}$ is at least $1 - 1/T$.

Consider the following two matrices.

$$\mu_1 = \begin{bmatrix} 2 & 3 \\ 1 & 1 \end{bmatrix} \qquad\qquad \mu_2 = \begin{bmatrix} 2 & 3 \\ 1 & 1 + \epsilon \end{bmatrix}$$

Define $P_1$ as the distribution of the full history $H_T$ when using algorithm ALG when $\mu^* = \mu_1$. Define $P_2$ as the equivalent distribution when $\mu^* = \mu_2$.

We will proceed according to the following proof sketch. First, we will show that if $\mu^* = \mu_1$, then ALG with constant probability allocates $\tilde{O}(T^{2/3})$ items of type 2 to player 2 (Lemma 18). We next upper bound the total variation distance between $P_1$ and $P_2$. When two distributions are sufficiently close in total variation distance, then an event that has constant probability under one distribution also has constant probability under the other distribution. Therefore, Lemma 18 and the closeness in TV distance of $P_1$ and $P_2$ together imply that if $\mu^* = \mu_2$, then ALG with constant probability allocates $\tilde{O}(T^{2/3})$ items of type 2 to player 2. When ALG allocates $\tilde{O}(T^{2/3})$ items of type 2 to player 2, then ALG cannot satisfy the envy-freeness constraints for $\mu_2$. Therefore, the previous two sentences together imply that with constant probability, ALG will not satisfy the envy-freeness constraints for all $t$ when $\mu^* = \mu_2$, which is a contradiction.

**Lemma 18.** *Using the notation defined above,*

$$\mathbb{E}_{P_1}[N_{22}^T] \leq T^{2/3}$$

*and*

$$\Pr_{P_1}\left(\sum_{t=0}^{T-1} X_{22}^t > \frac{T^{2/3}}{\log(T)}\right) < 1/8. \tag{37}$$

Intuitively, both equations in Lemma 18 bound how many times an item of type 2 is allocated to player 2. The first equation bounds in expectation while the second equation bounds elements of the fractional allocations $X^t$. The proof of Lemma 18 is given in Appendix G.1.

**Lemma 19.** *Using the notation from above,*

$$\text{KL}(P_1, P_2) = \mathbb{E}_{P_1}[N_{22}^T] \cdot \text{KL}\left(\mathcal{N}(1, 1), \mathcal{N}(1 + \epsilon, 1)\right)$$

*proof.* Define $f_{ik}^1$ as the probability density function (pdf) of $\mathcal{N}((\mu_1)_{ik}, 1)$ and $f_{ik}^2$ as the pdf of $\mathcal{N}((\mu_2)_{ik}, 1)$. We let $f_{\mathcal{N}(\mu, \sigma^2)}$ be the pdf of a normal distribution with mean $\mu$ and variance $\sigma^2$.

For any history $H_T = \{(k_t, i_t, v_t)\}_{t=0}^{T-1}$, recall that $X^t$ is the fractional allocation chosen by ALG at time $t$ given history $H_t$. Then we have that for any $H_T$

$$P_1(H_T) = \prod_{t=0}^{T-1} \left(\frac{1}{m} X_{i_t k_t}^t f_{i_t k_t}^1(v_t)\right),$$

The $1/m$ term comes from item $t$ having a $1/m$ probability of being of type $k_t$. The $X_{i_t k_t}^t$ is the probability that ALG allocates the item of type $k_t$ to player $i_t$ at time $t$, and finally $f_{i_t k_t}^1(v_t)$ is the probability of seeing value $v_t$ given that the item of type $k_t$ was allocated to player $i_t$. Similarly, we have that

$$P_2(H_T) = \prod_{t=0}^{T-1} \left(\frac{1}{m} X_{i_t k_t}^t f_{i_t k_t}^2(v_t)\right).$$

Therefore, we have that

$$KL(P_1, P_2) = \mathbb{E}_{H_T \sim P_1} \left[ \log \left( \frac{P_1(H_T)}{P_2(H_T)} \right) \right]$$

$$= \mathbb{E}_{H_T \sim P_1} \left[ \log \left( \frac{\prod_{t=0}^{T-1} \left( \frac{1}{m} X_{i_t k_t}^t f_{i_t k_t}^1(v_t) \right)}{\prod_{t=0}^{T-1} \left( \frac{1}{m} X_{i_t k_t}^t f_{i_t k_t}^2(v_t) \right)} \right) \right]$$

$$= \mathbb{E}_{H_T \sim P_1} \left[ \log \left( \frac{\prod_{t=0}^{T-1} \left( f_{i_t k_t}^1(v_t) \right)}{\prod_{t=0}^{T-1} \left( f_{i_t k_t}^2(v_t) \right)} \right) \right]$$

$$= \mathbb{E}_{H_T \sim P_1} \left[ \log \left( \prod_{t=0}^{T-1} \frac{f_{i_t k_t}^1(v_t)}{f_{i_t k_t}^2(v_t)} \right) \right]$$

$$= \mathbb{E}_{H_T \sim P_1} \left[ \log \left( \prod_{t:(i_t, k_t)=(2,2)} \frac{f_{\mathcal{N}(1,1)}(v_t)}{f_{\mathcal{N}(1+\epsilon,1)}(v_t)} \right) \right]$$

$$= \mathbb{E}_{H_T \sim P_1} \left[ \sum_{t:(i_t, k_t)=(2,2)} \log \left( \frac{f_{\mathcal{N}(1,1)}(v_t)}{f_{\mathcal{N}(1+\epsilon,1)}(v_t)} \right) \right]$$

$$= \sum_{t=0}^{T-1} \mathbb{E}_{H_T \sim P_1} \left[ \mathbb{1}_{(i_t, k_t)=(2,2)} \log \left( \frac{f_{\mathcal{N}(1,1)}(v_t)}{f_{\mathcal{N}(1+\epsilon,1)}(v_t)} \right) \right]$$

$$= \sum_{t=0}^{T-1} \mathbb{E}_{H_T \sim P_1} \left[ \mathbb{1}_{(i_t, k_t)=(2,2)} \mathbb{E} \left[ \log \left( \frac{f_{\mathcal{N}(1,1)}(v_t)}{f_{\mathcal{N}(1+\epsilon,1)}(v_t)} \right) \Big| \mathbb{1}_{(i_t, k_t)=(2,2)} \right] \right]$$

$$= \sum_{t=0}^{T-1} \mathbb{E}_{H_T \sim P_1} \left[ \mathbb{1}_{(i_t, k_t)=(2,2)} \mathbb{E}_{v \sim \mathcal{N}(1,1)} \left[ \log \left( \frac{f_{\mathcal{N}(1,1)}(v)}{f_{\mathcal{N}(1+\epsilon,1)}(v)} \right) \right] \right]$$

$$= \sum_{t=0}^{T-1} \mathbb{E}_{H_T \sim P_1} \left[ \mathbb{1}_{(i_t, k_t)=(2,2)} \cdot KL \left( \mathcal{N}(1,1), \mathcal{N}(1+\epsilon, 1) \right) \right]$$

$$= KL \left( \mathcal{N}(1,1), \mathcal{N}(1+\epsilon, 1) \right) \mathbb{E}_{H_T \sim P_1} \left[ \sum_{t=0}^{T-1} \mathbb{1}_{(i_t, k_t)=(2,2)} \right]$$

$$= KL \left( \mathcal{N}(1,1), \mathcal{N}(1+\epsilon, 1) \right) \mathbb{E}_{H_T \sim P_1} \left[ N_{22}^T \right].$$

$\square$

We can use Lemma 18 and Lemma 19 to bound the KL-divergence between $P_1$ and $P_2$ by

$$\mathrm{KL}(P_1, P_2) = \mathbb{E}_{P_1}[N_{22}^T] \cdot \mathrm{KL}\left( \mathcal{N}(1,1), \mathcal{N}(1+\epsilon, 1) \right) = \frac{\mathbb{E}_{P_1}[N_{22}^T]\epsilon^2}{2} \leq \frac{1}{2}. \tag{38}$$

We next need the following result from probability theory that is a consequence of the Bretagnolle-Huber inequality.

**Lemma 20.** *For any two probability distributions $p$ and $q$ defined on the same space and for any measurable event $F$ in this space,*

$$p(F^C) + q(F) \geq \frac{1}{2} e^{-\mathrm{KL}(p,q)}.$$

*proof.* For any probability distributions $p$ and $q$, we have by the Bretagnolle-Huber inequality that

$$d_{\mathrm{TV}}(p, q) \leq 1 - \frac{1}{2} e^{-\mathrm{KL}(p,q)}.$$

For any event $F$,

$$d_{\text{TV}}(p,q) \geq |p(F) - q(F)| \geq p(F) - q(F) = 1 - p(F^C) - q(F).$$

Combining the above equations gives that

$$p(F^C) + q(F) \geq \frac{1}{2}e^{-\text{KL}(p,q)}. \tag{39}$$

$\square$

Taking $p = P_1$, $q = P_2$ and $F = \left\{ \sum_{t=0}^{T-1} X_{22}^t \leq \frac{T^{2/3}}{\log(T)} \right\}$ in Lemma 20, we have by Equation (38) that

$$\Pr_{P_1}\left(\sum_{t=0}^{T-1} X_{22}^t > \frac{T^{2/3}}{\log(T)}\right) + \Pr_{P_2}\left(\sum_{t=0}^{T-1} X_{22}^t \leq \frac{T^{2/3}}{\log(T)}\right) \geq \frac{1}{2}e^{-\text{KL}(P_1,P_2)} \geq 1/4. \tag{40}$$

Combining Equation (37) with Equation (40) gives

$$\Pr_{P_2}\left(\sum_{t=0}^{T-1} X_{22}^t \leq \frac{T^{2/3}}{\log(T)}\right) \geq 1/8. \tag{41}$$

If $\sum_{t=0}^{T-1} X_{22}^t \leq \frac{T^{2/3}}{\log(T)}$, then there must exist some time $t$ such that $X_{22}^t \leq \frac{T^{-1/3}}{\log(T)}$. If $X_{22}^t \leq \frac{T^{-1/3}}{\log(T)}$, then player 2's envy in expectation at time $t$ for player 1 under $\mu_2$ (for sufficiently large $T$) is

$$X_{11}^t \cdot 1 + X_{12}^t(1+\epsilon) - X_{21}^t \cdot 1 - X_{22}^t(1+\epsilon) = (1 - X_{21}^t) \cdot 1 + (1 - X_{22}^t)(1+\epsilon) - X_{21}^t \cdot 1 - X_{22}^t(1+\epsilon)$$

$$= 2 + \epsilon - 2X_{21}^t - 2X_{22}^t(1+\epsilon)$$

$$\geq \epsilon - 2X_{22}^t(1+\epsilon)$$

$$\geq \epsilon - \frac{2T^{-1/3}(1+\epsilon)}{\log(T)}$$

$$= T^{-1/3} - \frac{2T^{-1/3}(1+T^{-1/3})}{\log(T)}$$

$$> 0,$$

implying that $X^t$ does not satisfy the envy-freeness in expectation constraints under $\mu_2$. Therefore, Equation (41) implies that

$$\Pr_{P_2}(\text{EFE for } \mu_2 \text{ not satisfied}) \geq \Pr_{P_2}\left(\sum_{t=0}^{T-1} X_{22}^t \leq \frac{T^{2/3}}{\log(T)}\right) \geq 1/8. \tag{42}$$

This contradicts the assumption that ALG satisfies the envy-freeness in expectation constraints for $\mu_2$ with probability at least $1 - 1/T$ when $\mu^* = \mu_2$.

$\square$

### G.1 Proof of Lemma 18

*proof.* Define $E$ as the event that the algorithm ALG satisfies the envy-free in expectation constraints for $\mu_1$ and has regret less than $\frac{T^{2/3}}{\log(T)}$ for $\mu^* = \mu_1$. By assumption, $\Pr_{P_1}(E) \geq 1 - 1/T$.

If $\mu^* = \mu_1$, then the social welfare maximizing envy-free allocation is

$$Y^{\mu_1} = \begin{bmatrix} 0 & 1 \\ 1 & 0 \end{bmatrix}.$$

Furthermore, a fractional allocation $X^t$ satisfies envy-freeness in expectation for $\mu_1$ only if

$$X_{21}^t + X_{22}^t \geq 1. \tag{43}$$

Therefore, for any $X^t$ that satisfies envy-freeness in expectation for $\mu_1$, the regret at time $t$ is

$$\langle Y^{\mu_1}, \mu^* \rangle_F - \langle X^t, \mu^* \rangle_F = 4 - (2X_{11}^t + 3X_{12}^t + X_{21}^t + X_{22}^t)$$

$$\begin{aligned} &= 4 - (2(1 - X_{21}^t) + 3(1 - X_{22}^t) + X_{21}^t + X_{22}^t) \\ &= -1 + X_{21}^t + 2X_{22}^t \\ &\geq X_{22}^t. \hspace{5cm} \text{[Equation (43)]} \end{aligned}$$

This implies that the regret when $\mu^* = \mu_1$ of ALG when ALG satisfies envy-freeness in expectation for $\mu_1$ is

$$T \cdot \langle Y^{\mu_1}, \mu^* \rangle_F - \sum_{t=0}^{T-1} \langle X^t, \mu^* \rangle_F \geq \sum_{t=0}^{T-1} X_{22}^t.$$

By definition, under event $E$, the regret of ALG for $\mu^* = \mu_1$ is at most $\frac{T^{2/3}}{\log(T)}$ and ALG satisfies the envy-freeness in expectation constraints for $\mu_1$. Therefore, the previous equation implies that under event $E$,

$$\sum_{t=0}^{T-1} X_{22}^t \leq \frac{T^{2/3}}{\log(T)}. \tag{44}$$

Recall that $N_{22}^T$ is the number of times item of type 2 is allocated to player 2. Equation (44) implies that

$$\mathbb{E}_{P_1}[N_{22}^T \mid E] = \mathbb{E}_{P_1}\left[\sum_{t=0}^{T-1} X_{22}^t \mid E\right] \leq \frac{T^{2/3}}{\log(T)}.$$

This implies that for sufficiently large $T$,

$$\begin{aligned} \mathbb{E}_{P_1}[N_{22}^T] &= \mathbb{E}_{P_1}[N_{22}^T \mid E] \Pr_{P_1}(E) + \mathbb{E}_{P_1}[N_{22}^T \mid \neg E] \Pr_{P_1}(\neg E) \\ &\leq \mathbb{E}_{P_1}[N_{22}^T \mid E] + T \cdot \frac{1}{T} \\ &\leq \frac{T^{2/3}}{\log(T)} + 1 \\ &\leq T^{2/3}. \end{aligned} \tag{45}$$

Equation (44) also implies that for sufficiently large $T$,

$$\Pr_{P_1}\left(\sum_{t=0}^{T-1} X_{22}^t \leq \frac{T^{2/3}}{\log(T)}\right) \geq \Pr_{P_1}(E) \geq 1 - \frac{1}{T} > 7/8. \tag{46}$$

Taking the complement of this equation proves Equation (37). $\qquad \square$

