# OpenReview forum: "Honor Among Bandits: No-Regret Learning for Online Fair Division"
_NeurIPS.cc/2024/Conference — NeurIPS 2024 spotlight_

### Official Review · Reviewer_V1uC · 2024-07-01

**Soundness:** 3
**Presentation:** 3
**Contribution:** 3
**Rating:** 7
**Confidence:** 4

**Summary:**

The paper studies online fair division of indivisible items, where items arrive one at a time, and each item must be permanently assigned to a player upon arrival. Player utilities are initially unknown and must be learned via bandit-style feedback: there are a finite number of types of goods, and each player’s value for a good of type k is drawn independently from a distribution D_k. The authors aim to maximize utilitarian welfare (the sum of utilities) while satisfying a hard constraint of either envy-freeness in expectation (EFE) or proportionality in expectation (PE).

EFE and PE are evaluated with respect to the randomized allocation on each time step. That is, before observing the type of the new item, the learner proposes a randomized allocation X_t which determines how the item will be allocated for each possible type. Each X_t is required to satisfy the fairness constraint (EFE and PE), not the cumulative realized allocation. I initially found this rather unintuitive, but the authors effectively justify this choice by showing that:
1. Any EFE algorithm achieves realized cumulative envy at most $\sqrt{T} \log(T)$, while optimal is at least $\sqrt{T}$.
2. EFE at each time step does not cause any welfare loss compared to only requiring EFE at the end.

The main result is a simple explore-then-commit algorithm which is EFE or PE while achieving regret $\tilde{O}(T^{2/3})$ wrt welfare.

Along the way, the authors prove that the EFE and PE constraints satisfy the following property: either all players receive the same allocation, or the constraint can be satisfied with slack. This novel result formalizes the intuition that the “hardest case” for fair division is when all agents have the same utilities. EFE/PE can trivially be satisfied in that case by the uniform allocation, and in all other cases, EFE/PE can be satisfied with slack.

**Strengths:**

The presentation is clear and accessible. Although some of the modeling choices felt initially unintuitive, the authors did an excellent job of justifying those choices, as discussed above. The main result is satisfying, both in terms of the simple explore-then-commit algorithm and the regret bound. Essentially, the authors successfully solved the problem they chose to study.

I found the either-uniform-allocation-or-slack property (and the result that EFE and PE satisfy it) to be quite interesting as well. This result captures an important intuition in a simple way. The authors argue that this property may be of independent interest, and I agree.

**Weaknesses:**

Although I think the problem is reasonable and meshes well with prior work, I question whether the mathematical model has any real-world relevance. The authors use a running example of a food bank with multiple branches, with the idea that unpredictable perishable donations must be allocated immediately and fairly among the branches. I have several problems with this analogy:
1. Surely food banks wish to allocate according to need without worrying that one branch might “envy” another if the second branch simply has higher need?
2. The paper assumes additive utility, and each branch’s utility would be extremely non-additive (no one needs infinite cereal).
3. It’s hard to imagine the donations being so unpredictable.
To be clear: I’m not claiming that a mathematical model must perfectly match reality in order to provide real-world insights. But this paper’s model seems only tenuously connected to the real-world scenario that is used as motivation. If the authors have further justification for the food bank analogy, I would be interested to hear it.

To me, the most compelling story in this paper actually centers on the either-uniform-allocation-or-slack property. The authors establish this intuitive but nontrivial property, and demonstrate how it can be applied in a bandit context. I can easily imagine this property being useful for other fair division problems that might have more real-world relevance.

This is more minor, but I think the citation below is quite relevant and should be discussed along with Benadè et al.

Hakuei Yamada, Junpei Komiyama, Kenshi Abe, and Atsushi Iwasaki. Learning fair division from bandit feedback. International Conference on Artificial Intelligence and Statistics, 2024.

**Questions:**

Do you agree with my criticisms of the food bank analogy? Is there a different real-world situation that you believe is captured by your mathematical model? More broadly, do you believe that your mathematical model provides real-world insights, and if so, what are they?

Below is feedback that you're welcome to take or leave. I don't expect responses, even though some are phrased as questions.
1. I would suggest naming Property 2 so that I don’t have to keep calling it the either-uniform-allocation-or-slack property :)
2. “This distribution is independent of both player i’s distributions for other item types and other players’ distributions.” Distributions can’t really be independent; I think this should state that samples from those distributions are independent.
3. “Intuitively, this is equivalent to assuming that players have non-zero values for all item types and that their values have a finite upper bound.” I don’t think this is quite right. Assuming non-zero values would just require mu_{ik}^* > 0. I would describe this assumption as saying that the distribution means are “well-conditioned”.
4. Definitions 1 and 2: mu_i^* is a vector, right? I don’t think this was ever explicitly defined. (Only mu^* and mu_{ik}^* were, I think.) The definitions are also missing “for all i”.
5. “proof” should be capitalized at the start of proofs
6. Why not just use gamma_0 in Property 2? Why have for all gamma < gamma_0?
7. Is it intentional for the “2” in C_{P2} to be a hyperlink to property 2?
8. Consider mentioning the actual value of (or a bound on) gamma_0 is for each property?
9. For readability in Algorithm 1, consider writing indicator variables as \mathbb{1}(stuff) instead of \mathbb{1}_{stuff}.
10. The notation “\mu in \hat{\mu} \pm \varepsilon” is non-standard to my knowledge. Consider “\mu in [\hat{\mu} - \varepsilon, \hat{\mu} + \varepsilon]”
11. I found lines 154-178 to be quite compelling and and also quite important. Consider making this its own subsection? And maybe even including some of the Theorem statements from Appendix B?
12. I really like lines 323-330.

**Limitations:**

The authors thoroughly discussed the technical limitations of their work, but did not discuss what I see as the greatest limitation: the weak motivation of the problem and the potential lack of real-world insights.

---

> ### Author Rebuttal · Authors · 2024-08-06
>
> Thank you for all of your helpful feedback! We will make sure to incorporate your suggestions into the final version. Below, we address your questions about the motivating example.
>
> > Do you agree with my criticisms of the food bank analogy? Is there a different real-world situation that you believe is captured by your mathematical model? More broadly, do you believe that your mathematical model provides real-world insights, and if so, what are they?
>
> While this paper is primarily a theoretical contribution, we do believe that the food bank analogy is a reasonable application, and we would like to provide some further justification for this use case.
>
> In online fair division, the food bank redistribution problem is a canonical example that has been studied extensively theoretically and in practice. In fact, a recent work [1] describes a partnership with a program in Indiana that redistributes "rejected truckloads of food". The program, known as "Food Drop", allocates 10,000+ lbs of rejected food per month to food banks.  As in our setting, the rejected food is an online and unpredictable process, and therefore this application requires an online algorithm. [1] focuses on practical algorithms that allocate the food fairly among the different food banks, and they use envy-freeness as the fairness metric. [1] does not discuss how to incorporate feedback/learning into the fair algorithms, which is one of our main contributions.
>
> > Surely food banks wish to allocate according to need without worrying that one branch might “envy” another if the second branch simply has higher need?
>
> In the aforementioned application of online fair division (which explicitly aims for envy-freeness), there are many locations that all have relatively equal need. Similarly, we are assuming that all players (food banks) start with the same amount of need, which can for example be achieved by first normalizing by number of customers. With that in mind, envy-freeness can be seen as preventing situations where the need of a food bank aren't met when it was possible to meet them. For example, if one food bank needs tomatoes and another doesn't, allocating the tomatoes to the former food bank wouldn't create envy, whereas allocating them to the latter (thereby not meeting the needs of the former) would.
>
> > The paper assumes additive utility, and each branch’s utility would be extremely non-additive (no one needs infinite cereal).
>
> This is an excellent point. Additive utility is a standard assumption for online fair division due to the difficulty of studying non-linear utilities — unfortunately, even in online fair division without learning, there is not much previous work analyzing non-additive utilities. Studying non-additive utilities in online fair division would certainly be an interesting problem. That being said, if donations are delivered at most once a week and all of the donations are used between deliveries, then additive utilities may actually be a reasonable assumption.
>
> > It’s hard to imagine the donations being so unpredictable.
>
> We expect that donations to food banks are pretty random; for example, grocery store donations depend on what is left at the end of the day, which is highly variable. However, there also exists an equivalent way to phrase our problem that does not involve unpredictable donations. Instead of receiving an item of a random type at each time as in our paper, we instead receive a (non-random) bundle of items at each time that has an equal amount of all item types. For example, instead of there being 1/2 chance of receiving a pound of apples and 1/2 chance of receiving a pound of oranges, we could instead have 100% chance of receiving 1 pound of apples and 1 pound of oranges, which can be allocated to the same or different players. This formulation is actually equivalent to our problem due to the linearity of expectation, and the exact same algorithms and results hold.
>
> We hope the above discussion helped clarify your concerns with the food bank motivating example. If you found this discussion helpful, we would also be happy to include more discussion of the application in the final paper.
>
>
> [1] Marios Mertzanidis, Alexandros Psomas, and Paritosh Verma. "Automating Food Drop: The Power of Two Choices for Dynamic and Fair Food Allocation." arXiv preprint arXiv:2406.06363 (2024).

---

> > ### Comment · Reviewer_V1uC · 2024-08-10
> >
> > I mostly find the authors' response compelling, and I will raise my score by 1 point under the assumption that the further justification for their mathematical model is incorporated into the final version of the paper. The most compelling point in my opinion is the existence of a real application that matches this mathematical model, and I would recommend the authors discuss Food Drop explicitly in the final version.
> >
> > I will briefly mention that I find this aspect of their response unconvincing
> > > Additive utility is a standard assumption for online fair division due to the difficulty of studying non-linear utilities
> >
> > The fact that the assumption is standard doesn't make it realistic. And the fact that non-linear utilities are challenging to analyze doesn't make the actual utilities additive.

---

> > > ### Author Response · Authors · 2024-08-10
> > >
> > > Your response is greatly appreciated. We're happy to follow your suggestion and discuss Food Drop.
> > >
> > > > The fact that the assumption is standard doesn't make it realistic. And the fact that non-linear utilities are challenging to analyze doesn't make the actual utilities additive.
> > >
> > > This is well taken. Not to belabor the point, but perhaps the argument we briefly mentioned about the frequency of allocations is more compelling. For example, in the food allocation data of Lee et al. (2019), there were a total of 1760 donations from 169 donors over the course of five months, and 277 organizations received donations. This means that organizations receive donations every 3-4 weeks on average, suggesting that the donations can largely be regarded as independent.
> > >
> > > Min Kyung Lee, Daniel Kusbit, Anson Kahng, Ji Tae Kim, Xinran Yuan, Allissa Chan, Daniel See, Ritesh Noothigattu, Siheon Lee, Alexandros Psomas, Ariel D. Procaccia: WeBuildAI: Participatory Framework for Algorithmic Governance. Proc. ACM Hum. Comput. Interact. 3(CSCW): 181:1-181:35 (2019)

---

> > > > ### Comment · Reviewer_V1uC · 2024-08-10
> > > >
> > > > I think the frequency argument is reasonable. More generally, I think some empirical work may be warranted to determine the extent to which utilities are in fact additive, since as you say, this is quite a common assumption in this area. I understand that that's beyond the scope of the current submission, of course.

---

### Official Review · Reviewer_MmvH · 2024-07-13

**Soundness:** 3
**Presentation:** 4
**Contribution:** 3
**Rating:** 8
**Confidence:** 3

**Summary:**

The authors study an online fair division problem where items arrive online and need to be assigned to agents to maximize welfare while satisfying one of envy-freeness or proportionality. The novel consideration in their model is that the item valuations are drawn from an unknown distribution. They approach this as both a learning problem as well as an allocation problem. They provide a simple algorithm with small regret compared to an algorithm that knew the distributions beforehand. They also prove several structural results regarding their model.

**Strengths:**

The model is quite interesting, as are their results. The proofs are involved, and might be of independent interest.

**Weaknesses:**

N/A

**Questions:**

Is it possible to achieve non-trivial regret in the setting where the valuations of the agents are not independent? In the motivating example of the food pantry, if a delivered food item is of bad quality, the agents might all lower their valuation of this item together.

---

> ### Author Rebuttal · Authors · 2024-08-06
>
> > Is it possible to achieve non-trivial regret in the setting where the valuations of the agents are not independent? In the motivating example of the food pantry, if a delivered food item is of bad quality, the agents might all lower their valuation of this item together.
>
> This is definitely an interesting extension. By linearity of expectation, we would expect that correlation between arms does not change the regret of the algorithm. Similarly, because the fairness constraints are linear, correlation between players would probably not affect the envy-freeness in expectation constraints. However, the correlated arms could potentially allow the algorithm to learn faster (have lower regret) if the algorithm can extract information from the correlation.

---

### Official Review · Reviewer_Mvzr · 2024-07-26

**Soundness:** 3
**Presentation:** 3
**Contribution:** 3
**Rating:** 7
**Confidence:** 4

**Summary:**

In this work, the authors consider the problem of maximising the total utility in online fair allocation settings, where:
(i) T items having types in set [m] arrive online in T distinct rounds; (ii) the value $V_{i}(t)$ that each agent $i$ assigns to each item $t\in [T]$, when its type is $k$, is independently drawn from a sub-Gaussian distribution with unknown mean $\mu_{i,k}^*$; (iii) for each item that arrives online, its type is known, and based on this information and previous observations, it must be irrevocably assigned to some agent; (iv) the value $V_{i}(t)$ of each item $t$  can only be observed after it has been assigned, and this information allows the estimation of the unknown probability distributions as $t = 1, \ldots, T$ varies.

The authors design an Explore-Then-Commit algorithm that maximises the total utility of the various players over $T$ rounds, with a regret of the order $\tilde{O}(T^{2/3})$, where the maximisation is constrained by achieving fairness properties (envy-freeness or proportionality) in expected value, with high probability.

**Strengths:**

-This paper introduces an interesting setting of online fair allocation, where techniques derived from bandit algorithms are applied to the case where valuations are distributed according to unknown distributions.
-The approaches considered in Lemmas 1 and 2, which characterise fairness properties robust to small uncertainties in the underlying distribution, are interesting and novel.
-The paper certainly required a considerable technical effort.

**Weaknesses:**

I believe that the results could be written and presented more effectively (see the major comments below). Another minor criticism is that, although the results provided in Lemmas 1 and 2 are quite novel and there has been a considerable technical effort in writing the complete version, the results obtained are somewhat expected within the field of bandit algorithms and follow standard approaches.

Major comments:

-It should be highlighted from the beginning that the allocations returned by the algorithm are envy-free in expectation, but with high probability (that is, not always).
-The matrix notation could sometimes be avoided in favour of explicitly listing the various constraints.
-It would be clearer to present the main algorithm right from the start and explain informally how the various properties come into play.
-The fact that the presented LP has infinite constraints is mentioned in the conclusion. I suggest providing more details in the technical section.

Minor comments:

-The acronyms efe and pe are not defined (despite clear from the context).
-Remark 1: specify that the rows different from i and i' are composed of zeros only.
-Claims of Lemma 1 and Lemma 2: satisfy -> satisfies.

**Questions:**

No questions.

**Limitations:**

Yes.

---

> ### Author Rebuttal · Authors · 2024-08-06
>
> > I believe that the results could be written and presented more effectively (see the major comments below). [...] -It should be highlighted from the beginning that the allocations returned by the algorithm are envy-free in expectation, but with high probability (that is, not always). -The matrix notation could sometimes be avoided in favour of explicitly listing the various constraints. -It would be clearer to present the main algorithm right from the start and explain informally how the various properties come into play. -The fact that the presented LP has infinite constraints is mentioned in the conclusion. I suggest providing more details in the technical section.
>
> Thank you for your helpful comments! We will try to incorporate your presentational comments into the final version of the paper to make the exposition clearer. We will make sure to explicitly clarify that the algorithms are envy-free with high probability (in fact, no algorithm can do better than that and still have sub-linear regret). We will also add more discussion of the LP with infinite constraints near the algorithm and make it clear that it is still possible to find the solution in polynomial time.

---

> > ### Comment · Reviewer_Mvzr · 2024-08-12
> > **Rebuttal**
> >
> > I have read the authors' response carefully, and I am satisfied with their reply.
> >
> > I strongly believe that the rationale behind the considered setting will be even more exhaustive in the revised version. Moreover, although the case of additive functions is more restrictive compared to general monotone functions, I do not think this is an issue, as this setting is one of the most studied in the field and is well established in the literature. However, I believe that the interest in this setting should not be justified by the fact that the more general setting is difficult or not possible in online learning. Finally, I think that the tight lower bound presented by the authors deserves to be mentioned in the new version (including the complete proof in the appendix), as it further justifies the approach obtained for the upper bound.
> >
> > In light of the above comments, I will increase my score by one point.

---

> > > ### Author Response · Authors · 2024-08-12
> > >
> > > Thank you very much for your comment. We appreciate your take-aways from the entire discussion and are happy to follow your suggestions.
> > >
> > > > In light of the above comments, I will increase my score by one point.
> > >
> > > Just in case this somehow fell through the cracks, we believe the change (from 6 to 7) isn't reflected in your current score.

---

### Author Rebuttal · Authors · 2024-08-06

We would like to thank all the reviewers for their time and comments! We respond to specific comments in the individual rebuttals below.

Shortly after submitting our paper, we found a simple lower bound which shows that for our setting, no algorithm can do better than the ones we presented. We plan to include this in the revision and we believe the new result will strengthen the paper. We include a brief discussion here in case the reviewers are interested, but please feel free to ignore this section if not.

A weakness of the current paper (as acknowledged in the discussion section) is that perhaps a more sophisticated algorithm such as UCB or Thompson Sampling would be able to achieve a better regret of $\tilde{O}(\sqrt{T})$. However, we are actually able to show that $\tilde{O}(T^{2/3})$ is the best that any algorithm can do with a very simple example, given below.

The example and informal proof is as follows. Suppose there are two item types and two players. In this case envy-freeness and proportionality are equivalent, and therefore we will focus on the former. Consider the following two value matrices (with rows = players, columns = item types).

$\mu_1 = \\begin{bmatrix}
        2 & 3 \\\\
       1 & 1
        \\end{bmatrix}$

$    \mu_2 = \\begin{bmatrix}
        2 & 3 \\\\
        1 & 1 +T^{-1/3}.
        \\end{bmatrix}$

We can show that no algorithm can with probability $1-1/T$ achieve regret of less than $\tilde{\Omega}(T^{2/3})$ and satisfy envy-freeness in expectation for both of these distributions. First, note that the expected social welfare maximizing allocation for $\mu_1$ is to give all items of type 1 to player 2 and all items of type 2 to player 1. On the other hand, any envy-free allocation for $\mu_2$ must give $\tilde{\Omega}(T^{-1/3})$ fraction of items of type 2 to player 2. This implies that if an algorithm is unable to distinguish between $\mu_1$ and $\mu_2$, then either the regret will be $\tilde{\Omega}(T^{2/3})$ for $\mu_1$ or the algorithm will not be envy-free for $\mu_2$.

Therefore, any algorithm that has regret of less than $\tilde{\Omega}(T^{2/3})$ and satisfies envy-freeness for both $\mu_1$ and $\mu_2$ must distinguish betwen $\mu_1$ and $\mu_2$. The only way to do this is to allocate at least $\tilde{\Omega}(T^{2/3})$ items of type $2$ to player 2. However, this will result in regret under $\mu_1$ of $\tilde{\Omega}(T^{2/3})$.

A more rigorous version of this argument using standard techniques for lower bounds in multi-armed bandit problems would be included in the final paper.

---

### Decision · Program_Chairs · 2024-09-25

**Decision:**

Accept (spotlight)

**Comment:**

The paper analyzes an interesting setting on online allocation constrained to return envy-free allocations. The setting is interesting and novel. The adopted techniques are also interesting and require non-trivial effort. Results are relevant. The presentation of the paper can be improved by a better organization of the content and a more detailed discussion about some of the assumption (see, e.g., additivity).